# Functional analysis of genetic variants in the high-risk breast cancer susceptibility gene *PALB2*

Rick A.C.M. Boonen[1], Amélie Rodrigue[2,3], Chantal Stoepker[1], Wouter W. Wiegant[1], Bas Vroling[4], Milan Sharma[1], Magdalena B. Rother[1], Nandi Celosse[1], Maaike P.G. Vreeswijk [1], Fergus Couch[5], Jacques Simard [2,6], Peter Devilee [1,7], Jean-Yves Masson [2,3] & Haico van Attikum [1*]

Heterozygous carriers of germ-line loss-of-function variants in the DNA repair gene *PALB2* are at a highly increased lifetime risk for developing breast cancer. While truncating variants in *PALB2* are known to increase cancer risk, the interpretation of missense variants of uncertain significance (VUS) is in its infancy. Here we describe the development of a relatively fast and easy cDNA-based system for the semi high-throughput functional analysis of 48 VUS in human *PALB2*. By assessing the ability of *PALB2* VUS to rescue the DNA repair and checkpoint defects in *Palb2* knockout mouse embryonic stem (mES) cells, we identify various VUS in *PALB2* that impair its function. Three VUS in the coiled-coil domain of PALB2 abrogate the interaction with BRCA1, whereas several VUS in the WD40 domain dramatically reduce protein stability. Thus, our functional assays identify damaging VUS in *PALB2* that may increase cancer risk.

---

[1] Department of Human Genetics, Leiden University Medical Center, Leiden 2333 ZC, The Netherlands. [2] CHU de Québec-Université Laval Research Center, Oncology Division, Québec City, QC G1R 3S3, Canada. [3] Department of Molecular Biology, Medical Biochemistry and Pathology, Laval University Cancer Research Center, Québec City, QC G1V 0A6, Canada. [4] Bio-Prodict, Nijmegen 6511 AA, The Netherlands. [5] Department of Laboratory Medicine and Pathology, Mayo Clinic, Rochester, MN 55905, USA. [6] CHU de Québec Research Center, Endocrinology and Nephrology Division, Québec City, QC G1V 4G2, Canada. [7] Department of Pathology, Leiden University Medical Center, Leiden 2333 ZC, The Netherlands. *email: h.van.attikum@lumc.nl

Germline loss-of-function (LOF) variants in the breast cancer susceptibility genes BRCA1 and BRCA2 are known to result in an approximately tenfold increased lifetime risk of developing breast cancer[1]. Similar to these genes, mono-allelic LOF variants in the gene encoding partner and localizer of BRCA2 (PALB2) also increase the risk of breast cancer[2], whereas bi-allelic LOF variants cause Fanconi anemia (FA)[3]. It is now well established that women who carry pathogenic variants in PALB2 are at a similar risk for breast cancer as those who carry pathogenic variants in BRCA2[1,4]. Therefore, PALB2 takes a valid place in breast cancer predisposition gene panel tests and is becoming widely included in breast cancer clinical genetics practice. This has already led to the identification of numerous variants in PALB2, which may associate with breast cancer (as of September 2019, 1301 PALB2 VUS have already been reported in ClinVar). However, current risk estimates for PALB2 variants have so far only been based on truncating variants that are predicted to fully inactivate the protein[5]. For most missense variants the impact on protein function is unclear and therefore the associated cancer risk is unknown. Assessment of pathogenicity of such variants of uncertain significance (VUS), therefore relies mostly on co-segregation with disease, co-occurrence with known pathogenic variants, and family history of cancer. To extend the utility of PALB2 genetic test results, additional methods for interpreting VUS are urgently required.

A key facet of interpreting VUS in PALB2 is understanding their impact on PALB2 protein function. PALB2 exists as oligomers that can form a complex with BRCA1 and BRCA2 and the recombinase RAD51[6,7]. This involves PALB2's N-terminal coiled-coil (CC) domain for interaction with BRCA1[7] and its C-terminal WD40 domain for interaction with BRCA2[8]. The PALB2-BRCA1/2-RAD51 complex plays an essential role in homologous recombination (HR), which is a critical pathway for the repair of highly-deleterious DNA double-strand breaks (DSBs). Following their detection, the ends of a DSB are resected to generate stretches of 3' single-stranded DNA (ssDNA), which are bound by the ssDNA-binding protein RPA. PALB2 becomes recruited to these resected DSB ends in a manner dependent on BRCA1 to facilitate the assembly of BRCA2 and RAD51 onto broken DNA ends. RAD51, in turn, catalyzes strand invasion and DNA transfer, usually from a sister chromatid available in S/G2 phase[6,7,9], ultimately leading to error-free repair of DSBs.

Germline nonsense and frameshift variants in BRCA1, BRCA2, and PALB2 give rise to a characteristic genome instability signature that is associated with HR deficiency[10]. Targeting this HR deficiency has proven to be effective in PARP inhibitor (PARPi)-based cancer treatment, during which the ensuing DSBs can be repaired by HR in healthy cells, but not in HR-deficient cancer cells[11,12]. While PARP inhibitor-based therapy holds great promise for the treatment of HR-deficient cancers, a major obstacle is that clinical testing of these tumors often reveals numerous VUS in BRCA1, BRCA2, and PALB2, for which the effect on HR and the response to PARP inhibitor-based therapy is often unclear.

For BRCA1 and BRCA2, functional assays that mostly use HR as a read-out have been established to assess the effect of VUS on protein function[13–17]. These assays have successfully determined the functional consequences and potential therapy response of a variety of VUS. However, with regard to PALB2, the functional analysis of variants is still in its infancy even though there is a clear clinical demand. Here, we fill this gap by describing the development of a robust functional assay for the analysis of VUS in PALB2. The assay allows a semi high-throughput analysis of VUS in human PALB2 cDNA in Palb2 knockout mouse embryonic stem (mES) cells using HR, PARPi sensitivity and G2/M checkpoint maintenance as read-outs. We identify at least 14 PALB2 VUS that strongly abrogate PALB2 function. Moreover,

PALB2 VUS located in the WD40 domain have a high tendency to impair PALB2 protein function by affecting its stability, whereas PALB2 variants located in the coiled-coil domain tend to impair its interaction with BRCA1. Thus, we report on the development of a relatively rapid and easy functional assay that can determine the functional consequences of VUS in PALB2, thereby facilitating cancer risk assessment and predicting therapy response.

## Results

**A cell-based functional assay for PALB2 variants.** For the analysis of PALB2 variants we envisioned a cell-based assay that allows for reliable semi high-throughput testing of variants in human PALB2. This cell-based approach should combine efficient integration and equal expression of human PALB2 cDNA carrying these variants in a cellular background devoid of endogenous Palb2 and with the ability to assess their effect on HR. To this end, we introduced the well-established DR-GFP reporter into IB10 mES cells, which are highly proficient in HR (Fig.1a, Supplementary Fig. 1a–c)[18]. The HR efficiency was nearly identical in all 3 correctly targeted clones (~10%) (Supplementary Fig. 1d) and clone 5 was selected for further experiments.

Next, we introduced the recombination-mediated cassette exchange (RMCE) system into cells from clone 5[13]. One component of this system, which consists of an acceptor cassette with F3 and Frt sites (Fig. 1a, Supplementary Fig. 2), was correctly integrated at the Rosa26 locus in 1 out of 6 targeted clones (Supplementary Fig. 2a,b). The other component is an exchange cassette that carries a promoterless neomycin selection marker and an EF1α promotor fused to human PALB2 cDNA flanked by F3 and Frt sites. This exchange cassette can be used for FlpO-mediated, site-specific integration of human PALB2 cDNA at the RMCE acceptor cassette (Fig. 1a)[19]. This would allow for stable expression of human PALB2, which we envisioned in a cellular background devoid of endogenous Palb2.

Since knockout (KO) of PALB2 is embryonic lethal[20–22], it has been notoriously difficult to generate PALB2^KO cells. However, since p53 deficiency could partially rescue in utero development of Palb2^KO mice, we decided to generate Palb2^KO mES cells in a p53-deficient background. In addition to facilitating the KO of Palb2, deficiency in both p53 and Palb2 may also mimic tumor settings, as somatic TP53 mutations are common in breast cancer associated with BRCA1/2[23,24] and PALB2[25]. We first employed CRISPR/Cas9-based genome editing to knockout mouse Trp53 in cells harboring DR-GFP and the RMCE acceptor cassette (Fig. 1a, b, Supplementary Fig. 3a,b). Subsequent analysis of 4 Trp53^KO clones revealed that HR remained unaffected in these cells (Fig. 1b), allowing functional analysis in this genomic background using HR as a read-out. Trp53^KO clone 3 had the highest percentage of cells (~50%) with a normal chromosome number (i.e., 40 chromosomes) (Fig. 1c) and was therefore selected for further experiments.

Finally, we applied CRISPR/Cas9-mediated genome-editing to knockout mouse Palb2 (Fig. 1d, Supplementary Fig. 3c, d). As expected, the efficiency of HR in the DR-GFP reporter assay was strongly reduced (by ~95%) in Trp53^KO/Palb2^KO cells when compared to that in Trp53^KO cells alone (Fig. 1d). To test whether human wild-type PALB2 can complement this defect, we stably expressed wild-type human PALB2 cDNA using RMCE (Fig. 1a). Importantly, due to site-specific integration, the promoterless neomycin gene will be driven by the endogenous Rosa26 promoter, which enhances targeting efficiency and allows for selection of integrants on medium containing neomycin. Indeed, we observed PALB2 expression in all individual neomycin-resistant clones that were tested for PALB2 expression

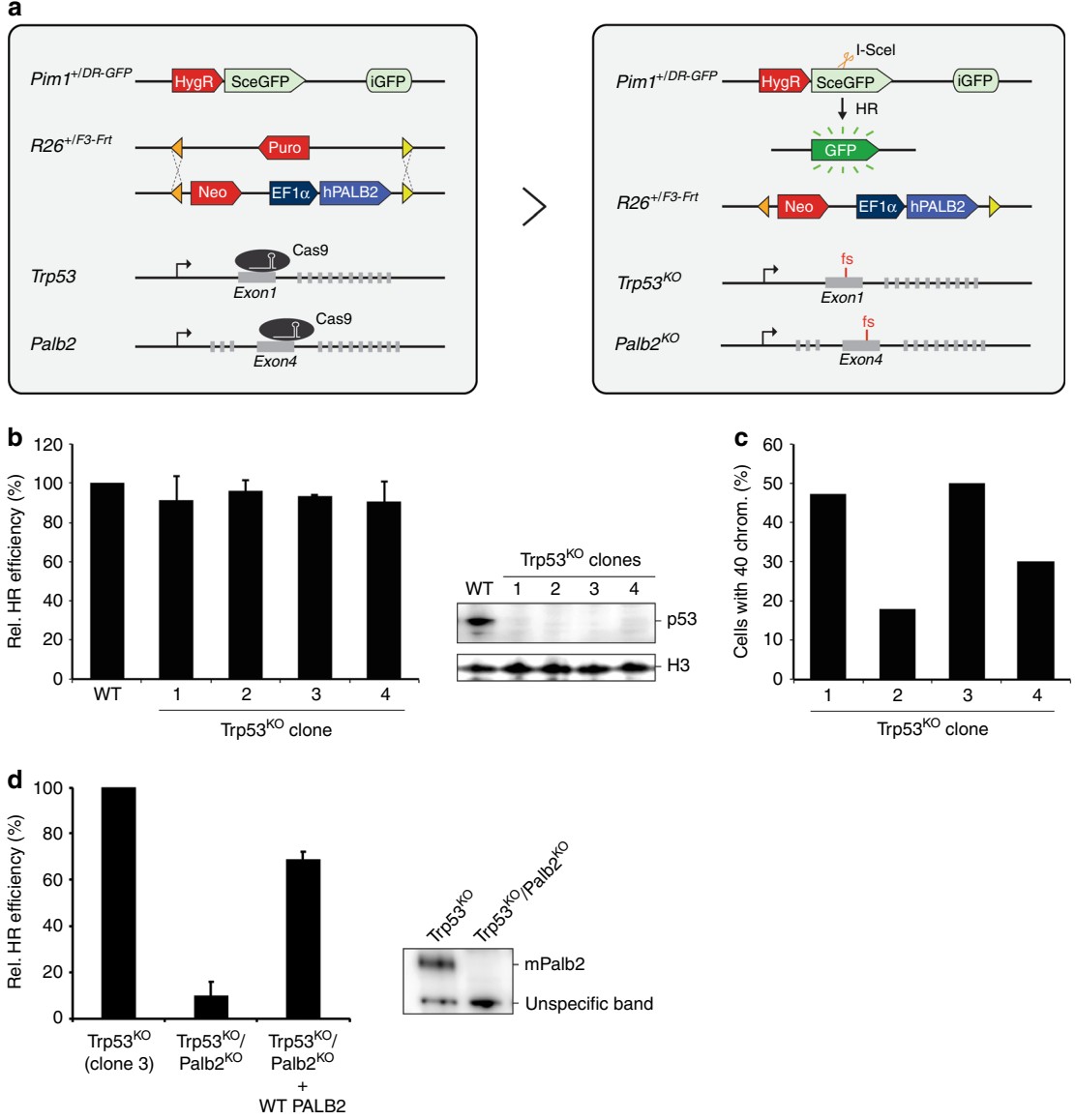

**Fig. 1** Development of a cDNA-based complementation system for the functional analysis of human *PALB2*. **a** Schematic of the cDNA-based complementation system for functional analysis of human *PALB2*. The DR-GFP reporter for HR and recombination-mediated cassette exchange system (RMCE) for site-specific integration and expression of a human *PALB2* cDNA were incorporated at the mouse *Pim1 and Rosa26* (*R26*) loci, respectively. Endogenous mouse *Trp53* was targeted with CRISPR/Cas9 using a gRNA against exon 1, whereas endogenous *Palb2* was targeted with a gRNA against exon 4 (left). Transient expression of the I-SceI endonuclease in *Trp53^KO^/Palb2^KO^* cells expressing human *PALB2* cDNA (with or without a variant) allows for assessment of the HR efficiency using the DR-GFP reporter (right). **b** DR-GFP assay in *Trp53^KO^* mES cell clones transfected with an I-SceI and mCherry co-expression vector. GFP expression was monitored by Fluorescence-Activated Cell Sorting (FACS). Data represent mean percentages (±SEM) of GFP-positive cells among the mCherry-positive cells relative to that for the wild type (WT), which was set to 100%, from two independent experiments (left). Western blot analysis of *Trp53* expression in 4 *Trp53^KO^* mES cell clones. Histone 3 (H3) was a loading control (right). **c** Karyotyping of *Trp53^KO^* mES clones from (**b**). The bar graph shows the percentages of cells with 40 chromosomes ($n = 50$ cells per condition). **d** DR-GFP assay in *Trp53^KO^* and *Trp53^KO^/ Palb2^KO^* mES cells expressing WT *PALB2* or not. Cells were transfected with an I-SceI and mCherry co-expression vector. GFP expression was monitored by FACS. Data represent mean percentages (±SEM) of GFP-positive cells among the mCherry-positive cells relative to that for *Trp53^KO^* cells, which was set to 100%, from four independent experiments (left). Western blot analysis of mouse (m)Palb2 expression in *Trp53^KO^* and *Trp53^KO^/Palb2^KO^* (clone 3) mES cells (right). An unspecific band was a loading control (right). Source data are provided as a Source Data file.

(Supplementary Fig. 4a). However, since some differences in *PALB2* expression were observed between single clones, we pooled the neomycin-resistant clones (~500 clones) prior to examining the HR efficiency (Supplementary Fig. 4b), ruling out any effects on HR caused by differences in *PALB2* expression. We found that HR was efficiently rescued (by ~68%) following expression of human *PALB2* in the *Trp53^KO^/Palb2^KO^* cells when compared to that in *Trp53^KO^* cells (Fig. 1d). Thus, we

have developed a highly efficient cDNA-based complementation system for the functional analysis of variants in human *PALB2*.

**Validation of a cell-based functional assay for *PALB2* variants.** To evaluate our system, we selected 12 truncating *PALB2* variants (Fig. 2a, red) that are known to be deleterious and associate with cancer and/or Fanconi anemia[3,4,26–28]. In addition, we selected 8

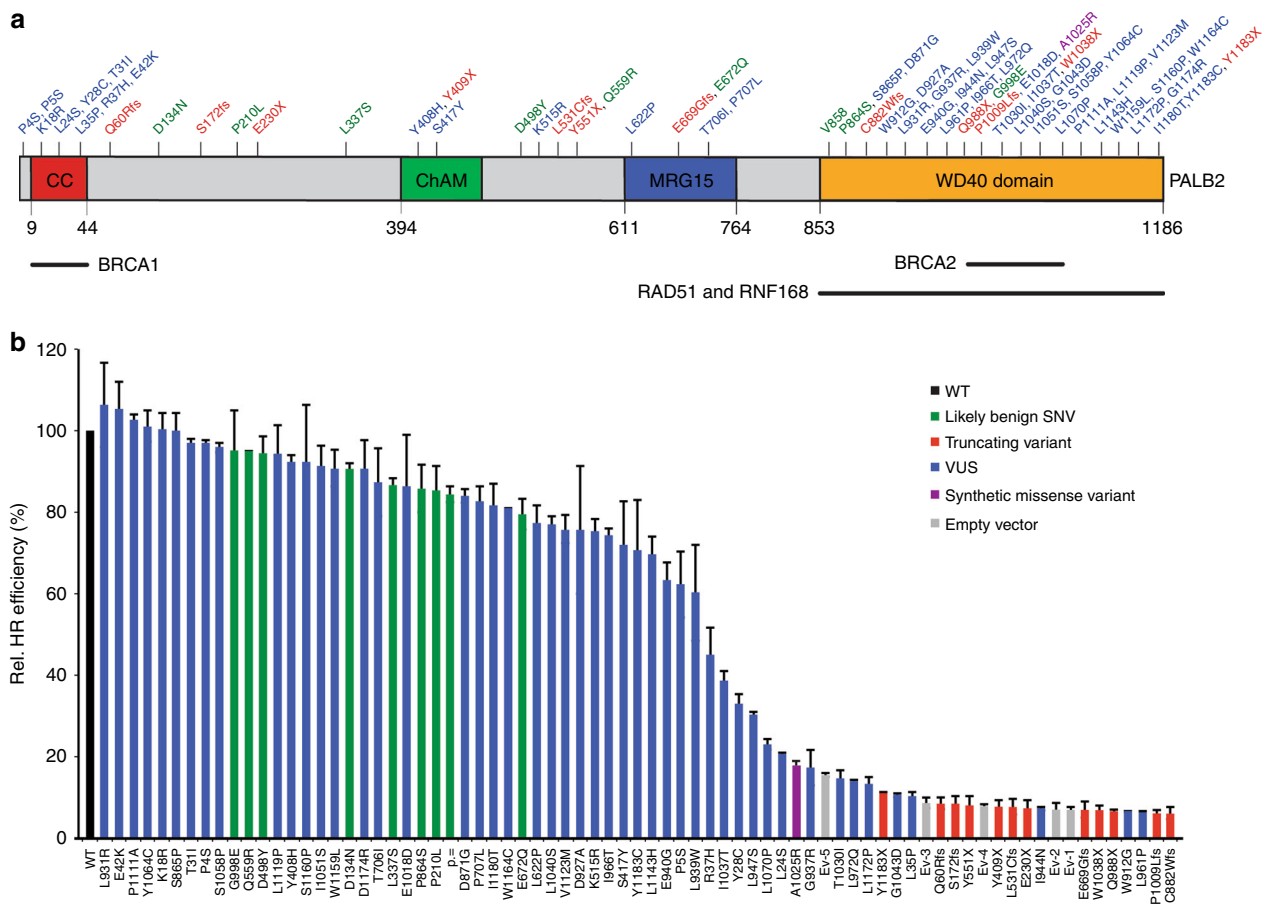

**Fig. 2** Human *PALB2* variants and their effect on HR. **a** Schematic representation of the PALB2 protein with variant positions indicated and categorized as either neutral (green), truncating (red), VUS (blue) and synthetic missense variant (purple). The amino acid numbers are shown to specify the evolutionarily conserved functional domains of PALB2. PALB2 regions involved in the interactions with BRCA1, BRCA2, RNF168, and RAD51 are indicated. **b** DR-GFP assay in *Trp53^KO^/Palb2^KO^* mES cells expressing human *PALB2* variants (or an empty vector control, Ev). Cells were transfected with an I-SceI and mCherry co-expression vector. GFP expression was monitored by FACS. Data represent mean percentages (±SEM) of GFP-positive cells among the mCherry-positive cells relative to wild type (WT), which was set to 100%, from two independent experiments, except for p.L939W and p.G998E for which data from three independent experiments are presented. Variants/conditions are categorized by color as either wild type (WT, black), likely benign SNV (green), truncating variant (red), VUS (blue), synthetic missense variant (purple) or empty vector (Ev, gray). Ev1–5 refer to Ev controls from 5 different replicates. Source data are provided as a Source Data file.

missense variants from the dbSNP database (Fig. 2a, green), which we expect to be benign/neutral because of their frequency in the general population (between 0.1 and 15% based on the 1000 Genomes Project). Site-directed mutagenesis was used to introduce these variants, as well as a synonymous variant (c.2574T>C, p. =), into the RMCE vector that carries human *PALB2* cDNA (Supplementary Data 1). Sequence-verified constructs were introduced by RMCE into the *Trp53^KO^/Palb2^KO^* mES cells, which were then subjected to DR-GFP assays. As expected, HR was dramatically reduced in cells carrying the empty vector (Ev) when compared to cells expressing human *PALB2* cDNA (i.e., a reduction in HR of ~90–95%) (Fig. 2b). Similarly, cells expressing human *PALB2* with a truncating variant displayed strong defects in HR. In contrast, cells that expressed either the benign/neutral variants or the synonymous variant showed HR levels comparable to that of cells expressing wild-type *PALB2* (Fig. 2b).

To corroborate these findings, we also examined whether cells expressing benign/neutral or truncating *PALB2* variants display sensitivity to PARPi treatment using a cellular proliferation assay. As expected, we found that *Trp53^KO^/Palb2^KO^* cells complemented with the Ev were hypersensitive to PARPi treatment when compared to those expressing wild-type human *PALB2* cDNA (Fig. 3a, Supplementary Figs. 5–7). Moreover, the expression of truncating *PALB2* variants led to a dramatically increased sensitivity to PARPi (at least by ~70%), while that of the benign/neutral variants did not (Fig. 3a, Supplementary Fig. 5). Thus, by measuring HR efficiencies using DR-GFP and PARPi sensitivity, our cell-based system reproducibly classifies benign/neutral and pathogenic/truncating variants based on their effect on PALB2 function in HR.

**Functional analysis of *PALB2* VUS.** In contrast to truncating variants in *PALB2*, the contribution of missense variants with respect to cancer risk is largely unclear. We, therefore, analyzed the effect of 48 *PALB2* VUS (Fig. 2a, *blue*) and one synthetic missense variant (p.A1025R) (Fig. 2a, *purple*)[29] on PALB2 function in HR. Many of these VUS have been identified during a multigene panel analysis for a large case-control association study performed by the BRIDGES consortium. In addition, several VUS were gathered from ClinVar (p.I944N, p.L24S and p.L1070P) and literature (p.K18R, p.Y28C, p.L35P, p.R37H) (Supplementary Data 1)[30,31]. Interestingly, we observed strong HR defects in

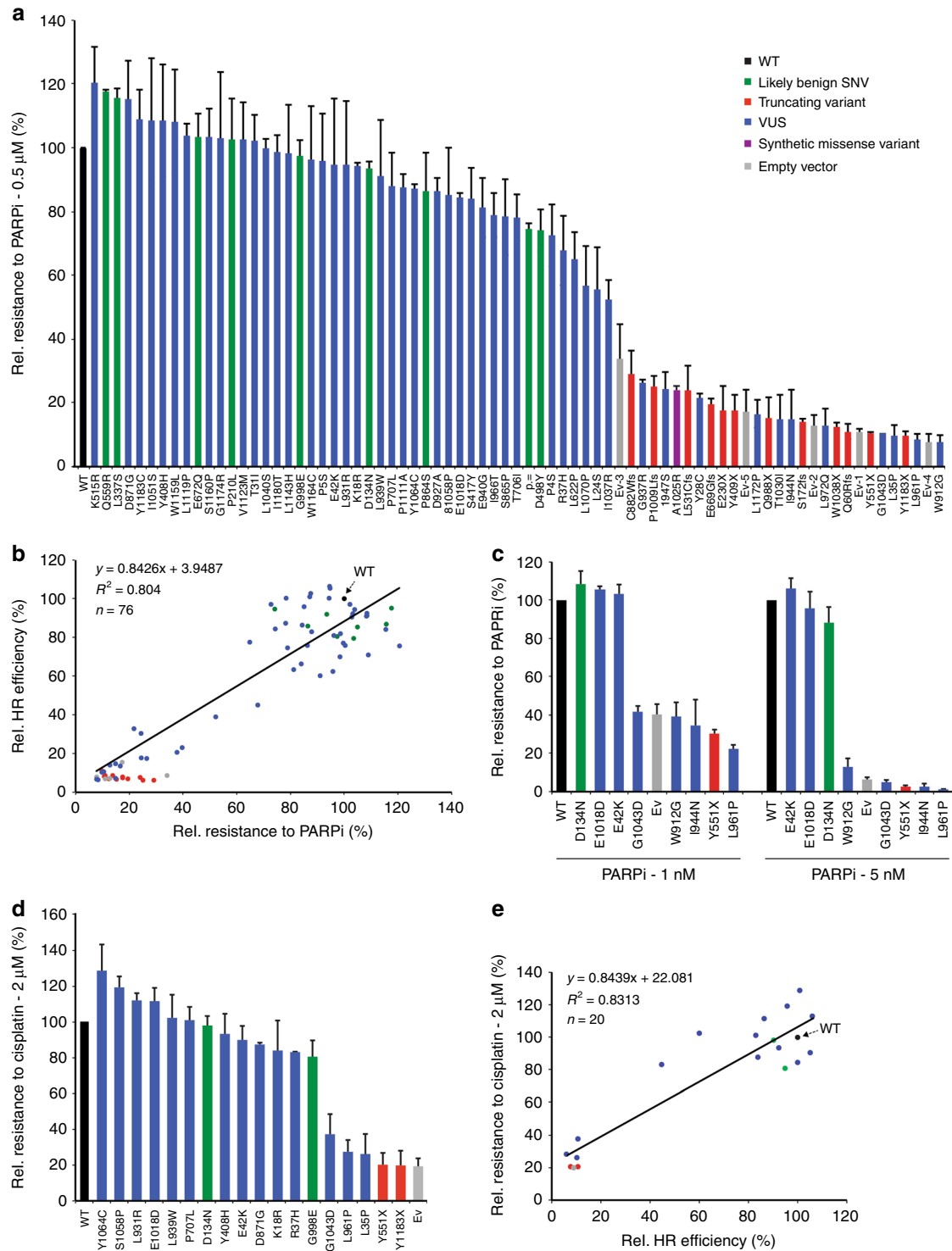

DR-GFP assays for p.L35P-, p.W912G-, p.I944N-, p.L961P-, p.G1043D-PALB2, exhibiting a ~90–95% reduction in HR, comparable to the truncating *PALB2* variants and the empty vector conditions (Fig. 2b). In addition, we also observed strong effects on HR for several other VUS (p.L24S, p.Y28C, p.G937R, p.L947S, p.L972Q, p.T1030I, p.I1037T, p.L1070P, and p.L1172P), as well as the synthetic missense variant p.A1025R in PALB2, reducing HR by ~60–90% when compared to wild-type PALB2 (Fig. 2b). A Fluorescence-Activated Cell Sorting (FACS)-based cell cycle analysis for 33 selected *PALB2* variants showed no effect on cell cycle distribution (Supplementary Fig. 8), excluding the

possibility that effects on HR were due to differences in cell cycle progression.

Next, we examined the effect of the 48 selected VUS and p.A1025R on PARPi sensitivity using a cellular proliferation assay. We observed that 11 VUS (p.Y28C, p.L35P, p.W912G, p.G937R, p.I944N, p.L947S, p.L961P, p.L972Q, p.T1030I, p.G1043D, and p.L1172P), as well as p.A1025R, displayed sensitivity to PARPi treatment comparable to that observed for *PALB2* truncating variants (Fig. 3a, Supplementary Figs. 6, 7). Importantly, when comparing the HR efficiency measured by DR-GFP and PARPi sensitivity assays, a strong positive correlation was observed for all

**Fig. 3** Functional analysis of *PALB2* VUS using PARP inhibitor and cisplatin sensitivity assays. **a** Proliferation-based PARP inhibitor (PARPi) sensitivity assay using *Trp53*^KO/*Palb2*^KO mES cells expressing human *PALB2* variants (or an empty vector control, Ev). Cells were exposed to 0.5 μM PARPi for two days. Cell viability was measured 1 day later using FACS. Data represent the mean percentage of viability relative to wild type (WT) (±SEM), which was set to 100%, from two independent experiments, except for p.P4S, p.P210L, p.L939W, and p.V1123M, for which data from three independent experiments is presented, and p.L24S and p.L1070P, for which data from four independent experiments is presented. Variants/conditions are categorized by color as in Fig. 2. **b** Scatter plot showing the correlation between HR efficiencies and PARPi sensitivity measured in Fig. 2b and Fig. 3a, respectively. Variants/conditions are categorized by color as in (**a**). The trendline indicates the positive correlation between the outcome of DR-GFP and PARPi sensitivity assays. **c** Clonogenic PARP inhibitor survival assay using *Trp53*^KO/*Palb2*^KO mES cells expressing human *PALB2* variants (or an empty vector control, Ev). Cells were exposed to the indicated concentrations of PARPi for 7–9 days after which surviving colonies were counted. Data represent the mean percentage of survival (±SEM) relative to cells expressing WT *PALB2*, which were set to 100%, from three independent experiments in case of treatment with 1 nM PARPi, and four experiments in case of treatment with 5 nM PARPi. Variants/conditions are categorized by color as in (**a**). **d** As in (**a**), except that cells were exposed to 2 μM cisplatin. Data represent the mean percentage of viability relative to WT (±SEM), which was set to 100%, from two independent experiments. **e** Scatter plot showing the correlation between HR efficiencies and cisplatin sensitivity measured in Fig. 2b and **d**. The trendline indicates the positive correlation between the outcome of DR-GFP and cisplatin sensitivity assays. Variants/conditions are categorized by color as in (**a**). Source data are provided as a Source Data file.

variants tested ($R^2 = 0.804$) (Fig. 3b). These results indicate that our complementary cell-based assays can determine the functional consequences of VUS in human *PALB2*. Most notably, taking the data from both assays into account, we identified at least 5 VUS (p.L35P, p.W912G, p.L961P, p.I944N, and p.G1043D) that affect PALB2 function to a similar extent as the truncating variants. The effect of these VUS on PARPi sensitivity was further evaluated using a clonogenic survival assay. This revealed that 4 *PALB2* VUS (p.W912G, p.L961P, p.I944N, and p.G1043D) also render cells hypersensitive to prolonged treatment with lower concentrations of PARPi (Fig. 3c). Consequently, such VUS may confer an increased cancer risk and serve as a target for PARPi-based therapy.

While PARPi treatment holds great promise for the treatment of HR-deficient tumors, an alternative strategy may be to treat with interstrand crosslink (ICL)-inducing chemotherapeutic drugs, since ICLs require HR for their repair[32]. We, therefore, analyzed several *PALB2* variants in their response to the ICL-inducing agent cisplatin. As expected, two truncating variants, p.Y551X and p.Y1183X, displayed strong sensitivity to cisplatin comparable to the empty vector condition (Fig. 3d). Consistent with the effects observed in the HR and PARPi assays, three *PALB2* VUS (p.L35P, p.L961P, and p.G1043D) were also sensitive to cisplatin. When comparing the HR efficiency measured by DR-GFP to cisplatin sensitivity, a strong correlation ($R^2 = 0.8313$) was observed (Fig. 3e). Thus, VUS in *PALB2* that impair HR may serve as targets for both PARPi- and ICL-based chemotherapy.

**Correlation of functional analysis and in silico prediction**. We next compared the outcome of our functional assays with the predictions of several in silico algorithms for all missense variants. For the prediction tools that give categorical results for missense variants, including PolyPhen[33], SIFT[34], and AlignGVGD[35], we observed little to no correlation with the outcome of DR-GFP and PARPi sensitivity assays (Supplementary Data 1). For instance, if we assume an HR efficiency of 40% or lower as damaging in the DR-GFP assay, then 24.1% of the missense variants (likely benign and VUS) are classified as damaging in our functional assay. However, we observed a gross overrepresentation of damaging variants when using PolyPhen (86.2%), SIFT (77.6%) and AlignGVGD (36.2%, counting C55 and C65). With respect to the latter, extreme caution should be taken as AlignGVGD classified at least two variants, which we found to be similarly damaging as truncating variants, as likely benign (p.W912G (C0) and p.I944N (C15); Supplementary Data 1). For in silico prediction tools that assign a continuous prediction score, such as (CADD[36] and REVEL[37]), we similarly observed a poor correlation with the outcome of DR-GFP and

PARPi sensitivity assays (Supplementary Fig. 9). For instance, based on cut-offs of 0.0–0.5 for benign variants and 0.5–1.0 for damaging variants, REVEL would only categorize three of the PALB2 VUS (p.D871G, p.W912G, and p.L931R) as damaging. However, both p.D871G and p.L931R appear to be fully functional in our assays. Thus, while REVEL severely underestimates the effects of VUS on protein function, it may also lead to false-positive predictions. Based on these observations, we conclude that predictive algorithms, as opposed to our functional analysis, are poor in predicting the effect of VUS on PALB2 protein function.

**VUS in the PALB2 WD40 domain affect protein stability**. Having identified *PALB2* variants that affect HR, we sought to address their mechanism of action. To this end, we first examined their effect on PALB2 expression by western blot analysis. For all benign variants, PALB2 expression was comparable to that of wild-type PALB2 (Fig. 4a). Similarly, most truncating and missense variants were unaffected in their expression levels, although the truncating variants resulted in the expression of the expected smaller proteins. However, for some truncating variants (p.Q899X, p.P1009Lfs, p.W1038X and p.Y1183X) and VUS located in the C-terminal WD40 domain (p.W912G, p.G937R, p.I944N, p.L947S, p.L961P, p.L972Q, p.T1030I, p.I1037T, p.G1043D, p.L1070P, and p.L1172P), low levels of expression were observed (Fig. 4a, red asterisk). Reverse transcription-quantitative (RT-qPCR) analysis indicated that these variants did not affect expression at the mRNA level (Fig. 4b). This suggests that the low abundance of PALB2 protein is likely the result of protein misfolding and/or instability.

Crystal structure studies of the PALB2 C-terminal WD40 domain suggested that loss of the last 3 amino acids of PALB2 caused by the FA-associated p.Y1183X variant disrupts the hydrogen bonding in the seventh blade of the WD40 domain[3,29]. Consistently, we also observed strongly reduced expression of PALB2 carrying this variant (p.Y1183X) (Fig. 4a). Thus, p.Y1183X may lead to in an incompletely folded PALB2 protein that is likely to be degraded rapidly. As such, it is not surprising that other truncating variants in the WD40 domain result in expression of a truncated protein that is unstable and degraded quickly. However, truncating PALB2 variants that lack the entire WD40 domain (p.E230X, p.Y409X, p.L531Cfs, p.Y551X, and p.E669Gfs) appeared to express well (Figs. 2a and 4a). Nevertheless, they have likely lost all of their ability to interact with BRCA2 and RAD51, thereby impairing HR completely. Consistently, we observed almost no difference in the extent to which the different truncated forms of PALB2 affect HR (Fig. 2b).

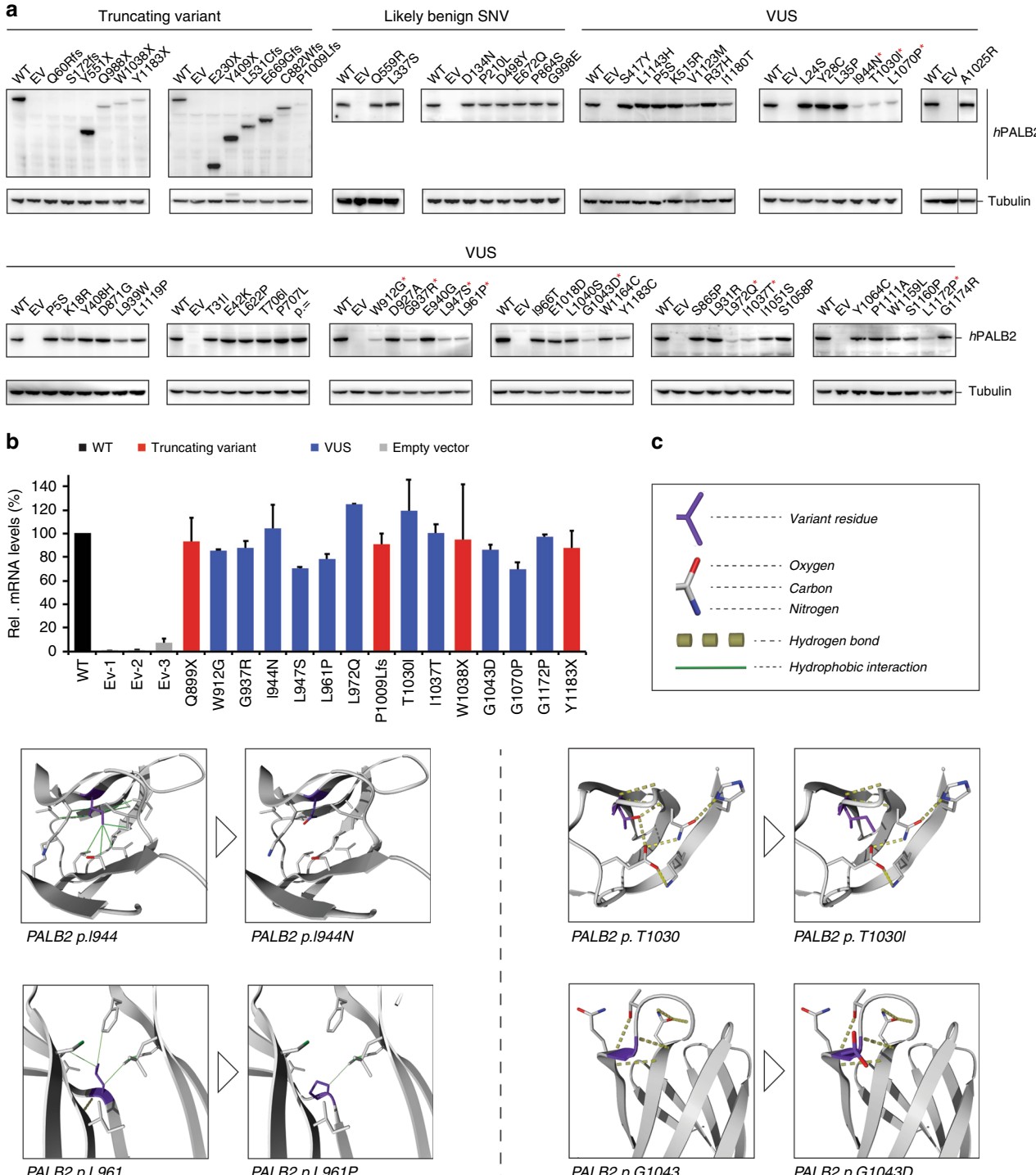

**Fig. 4** Effect of *PALB2* variants on protein expression and/or stability. **a** Western blot analysis of the expression of human *PALB2* variants in *Trp53*^KO/ *Palb2*^KO mES cells using an antibody directed against the N-terminus of PALB2. Wild-type (WT) human PALB2 and empty vector (Ev) served as controls on each blot. Tubulin was a loading control. Marked PALB2 variants (red *) showed low levels of protein expression. **b** RT-qPCR analysis of human *PALB2* variants from (**a**) with low expression levels (red *). Primers specific for human *PALB2* cDNA and the *Pim1* control locus were used. Data represent the mean percentage (±SEM) of *PALB2* mRNA relative to WT, which was set to 100%, from two independent RNA isolation experiments. Variants/conditions are categorized by color as either WT (black), truncating variant (red), VUS (blue) or empty vector (Ev, grey). Ev-1, -2, -3 refer to Ev controls from three different replicates. **c** Partial structures of the PALB2 WD40 domain showing the effect of 4 PALB2 variants exhibiting low protein expression as shown in (**a**). Partial structures without and with variant are shown side by side for each variant, indicating loss of stabilizing interactions (but not any possible conformational changes). Source data are provided as a Source Data file.

Our results suggest that the WD40 domain of PALB2 is extremely sensitive to variants that affect protein folding and/or stability. Using the crystal structure of the WD40 domain (2W18)[29], in silico modeling of all PALB2 VUS that display low expression levels indeed showed that all these amino acid substitutions are extremely unfavorable for correct folding of this domain. Starting with p.I944N, we see that this isoleucine is a well-conserved hydrophobic residue that is located in an antiparallel β-sheet and whose side-chain is part of a tightly packed hydrophobic environment (Fig. 4c). Replacement of this isoleucine with an asparagine will lead to the loss of stabilizing hydrophobic interactions due to the energetically unfavorable presence of a hydrophilic residue in a very hydrophobic environment. These opposed effects may destabilize the local environment and/or lead to folding problems. Comparable effects are predicted for p.L947S, p.L972Q, and p.I1037T (Supplementary Fig. 10). p.L961 is another example of a residue that is located in a β-sheet and is involved in several hydrophobic interactions (Fig. 4c). When it changes into a proline (p.L961P), all of these local interactions are lost. Furthermore, proline is unfavored, because it results in the loss of a backbone hydrogen bond, thereby destabilizing the associated β-sheet. Comparable effects are predicted for p.W912G, p.L1070P, and p.L1172P (Supplementary Fig. 10). However, for p.W912G the change into a very small glycine is also thought to result in excess flexibility at a position where this is not desired.

The side-chain of the hydrophilic residue p.T1030 is involved in an extensive network of hydrogen bonds and electrostatic interactions that extend across all 4 strands of the associated β-sheet (Fig. 4c). This variant will impair the formation of hydrogen bonds, as isoleucine is not capable of forming these bonds through its sidechain. Consistent with our findings (Fig. 4a), an earlier study also reported protein instability for p.T1030I[31]. Finally, p.G937 and p.G1043 are examples of glycine residues that provide structural flexibility at the beginning of a loop structure (Fig. 4c, Supplementary Fig. 10). Changing these residues into a larger and charged arginine (p.G937R) or aspartate (p.G1043D), will lead to deformation of the loop structure and probable loss of surrounding hydrogen bonds in the case of p.G1043D. Altogether, this in silico modeling may provide explanations for how these PALB2 VUS affect protein stability/expression levels. Nonetheless, some VUS for which similar destabilizing effects are predicted (p.D871G, p.L931R, p.E1018D, and p.W1164C) are fully functional in our HR-based assays, underpinning the importance of functional analysis of VUS.

**VUS in the PALB2 CC-domain disrupt the interaction with BRCA1.** In addition to the damaging VUS in PALB2's WD40 domain, we also found 4 PALB2 VUS (p.L24S, p.Y28C, p.L35P, and p.R37H) exhibiting strong effects on HR and PARPi sensitivity (Figs. 2a and 3a, Supplementary Fig. 6). These variants were all located in PALB2's N-terminal CC domain, which is required for interaction with BRCA1 (Fig. 2a)[6,9]. Indeed, the previously reported p.Y28C and p.L35P variants affected HR by impairing the interaction with BRCA1[30]. However, exactly how p.L24S and p.R37H impact HR is unclear, also because p.R37H has previously been reported not to affect the PALB2-BRCA1 interaction[30]. To examine this further, we transiently expressed YFP-tagged PALB2 carrying p.L24S, p.L35P or p.R37H in U2OS cells and performed pull-downs using GFP Trap beads. p.L24S, similar to p.L35P, failed to co-precipitate any endogenous BRCA1, whereas p.R37H partially affected the co-precipitation of BRCA1 (Fig. 5a). Additionally, we examined whether these VUS have an impact on the BRCA1-dependent localization of PALB2 to sites of DNA damage. To this end, YFP-tagged PALB2 carrying

p.L24S, p.L35P or p.R37H were transiently expressed in U2OS cells and examined for their localization at DNA damage-containing tracks generated by laser micro-irradiation. We found that all three VUS impaired the recruitment of PALB2 to sites of DNA damage (Fig. 5c, d). The effect of these VUS on PALB2's interaction with BRCA1 and localization at sites of DNA damage are highly consistent with the observed HR defect (Figs. 2b and 3a, Supplementary Fig. 6). Taken together, we identified p.L24S and R37H as VUS that impair PALB2's function in HR by abrogating its interaction with BRCA1, and consequently its BRCA1-dependent recruitment to DNA damage sites.

**PALB2 VUS affect G2/M-phase progression after DNA damage.** While PALB2 is essential for HR, two independent genetic screens identified *PALB2* as a critical regulator of the DNA damage-induced G2/M checkpoint response[38,39]. Another study demonstrated that *PALB2* plays a role in maintaining a proper G2/M checkpoint response in human cancer cells exposed to ionizing radiation (IR)[40]. Therefore, we addressed if VUS in *PALB2* would affect the DNA damage-induced checkpoint by measuring the mitotic fraction of $Trp53^{KO}$ and $Trp53^{KO}/Palb2^{KO}$ mES cells following exposure to IR. 1 h after exposure to 3 or 10 Gy of IR, both $Trp53^{KO}$ and $Trp53^{KO}/Palb2^{KO}$ mES cells showed an almost complete loss of mitotic cells, indicating efficient activation of the G2/M checkpoint in both cell types (Fig. 6a). While at 6 h after 3 Gy of IR the mitotic fraction of both $Trp53^{KO}$ and $Trp53^{KO}/Palb2^{KO}$ mES cells dramatically increased, we only observed this increase in $Trp53^{KO}/Palb2^{KO}$ mES after exposure to 10 Gy (Fig. 6a). Thus, *PALB2* is also required for the maintenance of the IR-induced G2/M checkpoint in mES cells.

This prompted us to assess the effect of 19 different *PALB2* variants on G2/M checkpoint maintenance. We expressed these variants, which were selected based on their differential impact on HR (Fig. 2a), in $Trp53^{KO}/Palb2^{KO}$ mES cells and determined the mitotic fraction 6 h after exposure to 10 Gy of IR. Importantly, expression of wild-type human *PALB2* rescued the G2/M checkpoint maintenance defect observed in $Trp53^{KO}/Palb2^{KO}$ mES cells, whereas expressing the empty vector or either of two truncating variants (p.Y551X and p.Y1183X) resulted in a checkpoint defect (Fig. 6b). Two benign variants (p.D134N and p.G998E) and 9 different VUS (p.K18R, p.E42K, p.Y408H, p.P707L, p.D871G, p.L931R, p.1018D, p.Y1046C, and p.S1058P) that did not impair HR, also did not impact the maintenance of the IR-induced G2/M checkpoint. In contrast, strong defects in G2/M checkpoint maintenance were observed for 3 VUS (p.L35P, p.L961P, and p.G1043D) and the synthetic missense variant p. A1025R that also abrogated HR (Fig. 2b), whereas p.R37H and p. L939W exhibited a moderate effect (Fig. 6b), consistent with their mild impact on HR (Fig. 2b). Accordingly, we found a strong correlation between the impact of *PALB2* variants on HR and G2/ M checkpoint maintenance ($R^2 = 0.8577$) (Fig. 6c). Interestingly, p.L35P and p.A1025R have been shown to abrogate the interaction of PALB2 with BRCA1 (Fig. 5a)[30] and BRCA2[29], respectively. This indicates that both the interaction with BRCA1 and BRCA2 is crucial for PALB2's function in controlling G2/M-phase progression following DNA damage, which is in accordance with observations in human cancer cells[40].

**Functional analysis of *PALB2* VUS in human cell-based assays.** To validate results from our mES cell-based assays, we selected 5 LOF VUS located in the WD40 domain of PALB2 (p.W912G, p. G937R, p.L947S, p.L961P, and p.G1043D) and tested their effect on HR in human cell-based assays. To this end, we first employed the CRISPR-LMNA HR assay, which monitors the integration of mRuby, into the Lamin A/C locus (LMNA) by CRISPR/Cas9-

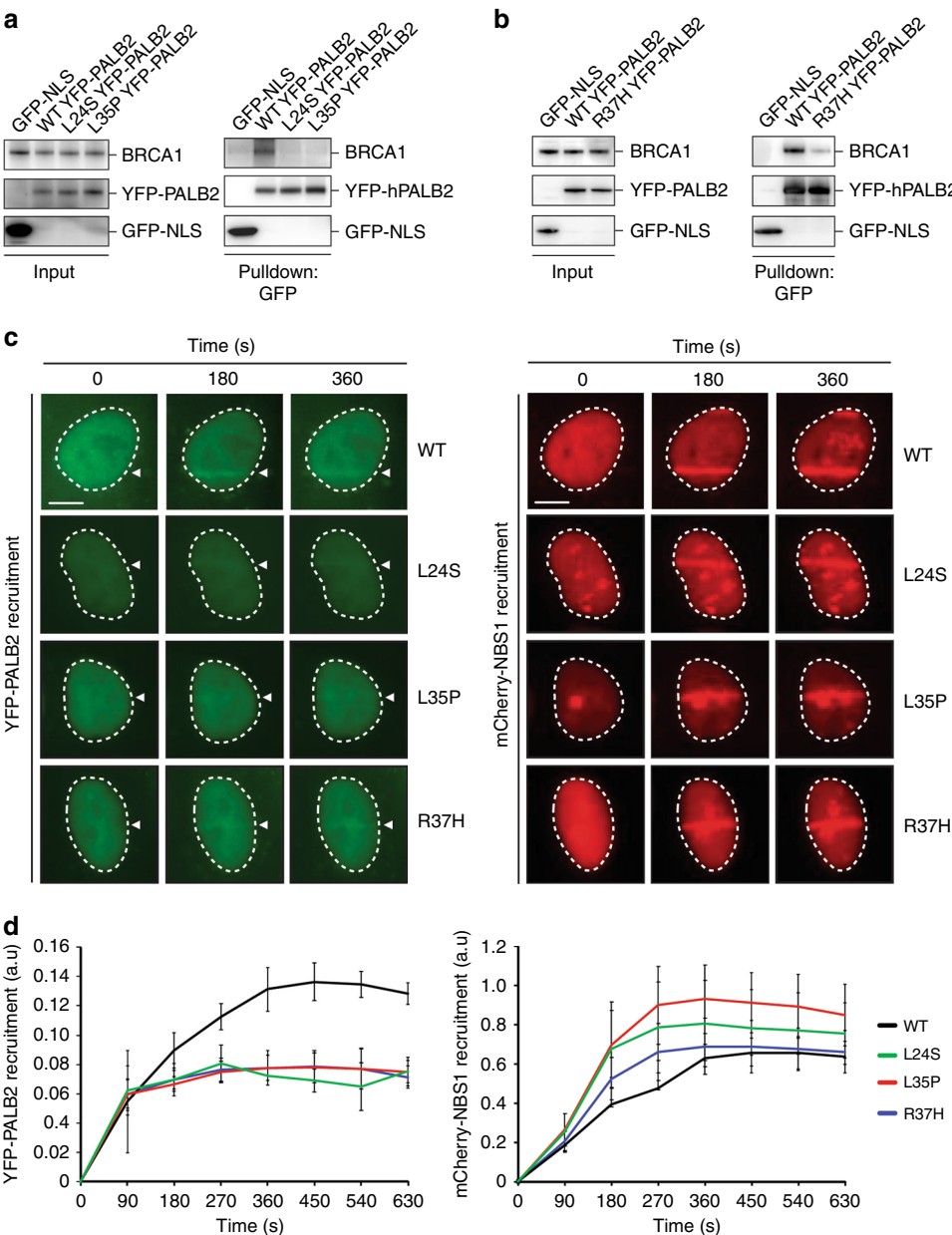

**Fig. 5** Effect of PALB2 VUS on the BRCA1 interaction and recruitment to DNA damage sites. **a** YPF/GFP pulldowns of the indicated proteins following transient expression in U2OS cells. GFP-NLS and YFP-PALB2-L35P served as negative controls. Western blot analysis was performed using antibodies against GFP and BRCA1. **b** As in (**a**), except for p.R37H. **c** Live cell imaging of the recruitment of the indicated YFP-PALB2 proteins to DNA damage tracks generated by laser micro-irradiation in U2OS cells. mCherry-NBS1, which was co-expressed with the indicated YPF-PALB2 proteins, served as a DNA damage marker. Representative images are shown. White triangles indicate irradiated regions. Scale bars: 5 μm. **d** Quantification of the recruitment of the indicated YFP-PALB2 proteins and mCherry-NBS1 to DNA damage tracks in cells from (**c**). Data represent the mean values (±SEM) from three independent experiments. Source data are provided as a Source Data file.

mediated HR (Supplementary Fig. 11a, b)[41]. Following siRNA-mediated knockdown of *PALB2* in U2OS cells, plasmids encoding the mRuby2-LMNA donor, Cas9 and a *LMNA* gRNA, and siRNA-resistant YFP-PALB2 with or without VUS, were co-transfected into these cells (Supplementary Fig. 11c). Four PALB2 VUS (p.W912G, p.G937R, p.L961P, and p.G1043D) showed a dramatic impact on the HR-mediated integration of mRuby (Fig. 7a). One VUS (p.L947S), had a moderate effect, although this is likely explained by the slightly higher transient expression of this variant (Supplementary Fig. 11c). We then assessed whether these VUS would affect PARPi sensitivity. To this end, siRNA-resistant YFP-PALB2 constructs carrying these VUS were

expressed in PALB2-depleted HeLa cells (Supplementary Fig. 11d). Four PALB2 VUS (p.W912G, p.G937R, p.L961P, and p.G1043D), showed a dramatic increase in PAPRi sensitivity, while 1 VUS (p.L947S) had a more moderate effect, consistent with findings from the CRISPR-LMNA HR assay (Fig. 7b). Altogether, these results corroborate our findings from the DR-GFP and PARPi sensitivity assays in mES cells (Figs. 2a and 3a, Supplementary Fig. 6).

Finally, PALB2 drives HR by promoting the accumulation of RAD51 at DSB sites. To further assess the impact of the 5 selected VUS on PALB2, we examined whether they affected the accumulation of RAD51 at IR-induced DSBs by measuring the

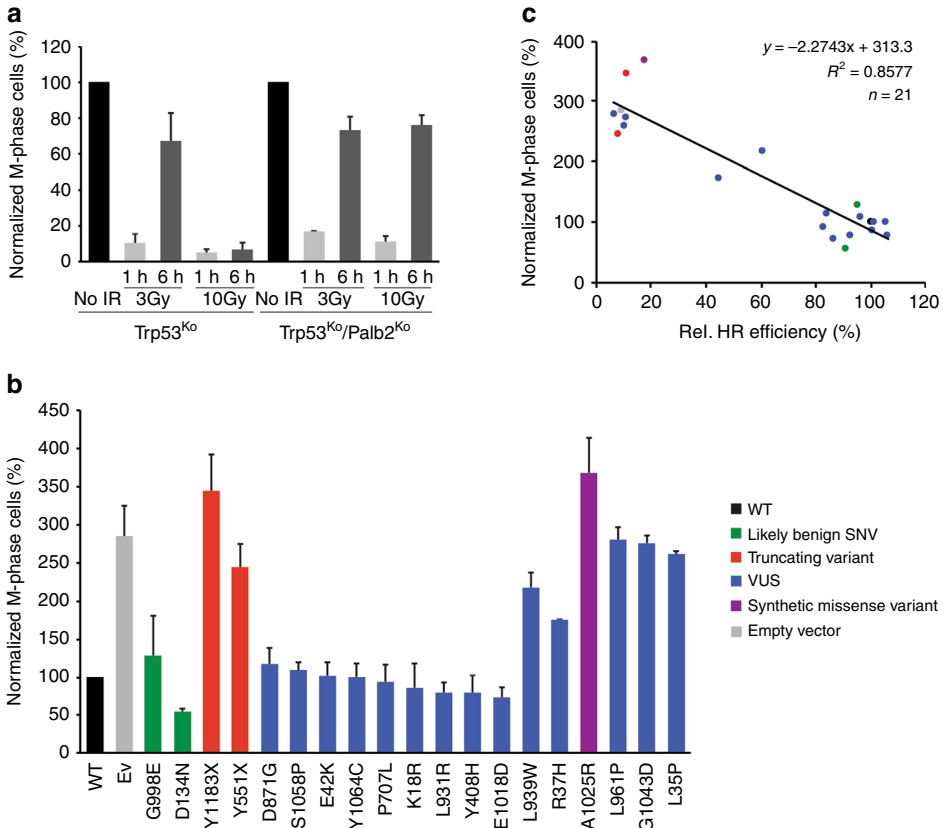

**Fig. 6** Effect of *PALB2* variants on the DNA damage-induced G2/M checkpoint. **a** *Trp53^KO/Palb2^KO* mES cells were irradiated with 3 or 10 Gy of IR and collected at the indicated time points after radiation exposure to assess the mitotic index by phospho-histone H3 (Ser10) staining and FACS analysis. Data represent the mean percentage of mitotic cells (±SEM) relative to the unirradiated cells (no IR), which was set to 100%, from two independent experiments. **b** *Trp53^KO/Palb2^KO* mES cells expressing the indicated *PALB2* variants were irradiated with 10 Gy of IR and collected 6 h after radiation exposure to assess the mitotic index by phospho-histone H3 (Ser10) staining and FACS analysis. For each variant, the mean percentage of mitotic cells (±SEM) from two independent experiments is shown relative to unirradiated cells, except for p.L939W and p.G998E for which data from three independent experiments is presented. Variants/conditions are categorized by color as in Fig. 2. **c** Scatter plot showing the correlation between the HR efficiencies and the mitotic index after IR as measured in Fig. 2b and **b**, respectively. Variants/conditions are categorized by color as in (**b**). The trendline indicates the negative correlation between the HR efficiency and mitotic index after IR, revealing a strong positive correlation between the impact of *PALB2* variants on HR and G2/M checkpoint maintenance. Source data are provided as a Source Data file.

formation RAD51 foci. HeLa cells were treated with siRNAs against endogenous *PALB2* and complemented by transient expression of siRNA-resistant YFP-PALB2, with or without VUS. Following exposure to IR, the average number of RAD51 foci was scored in cyclin-A- and YFP-PALB2-expressing S-phase cells (Fig. 7c, d). While 3 VUS (p.W912G, p.L961P, and p.G1043D) had a dramatic impact on the percentage of cells showing RAD51 foci, 2 VUS (p.G937R and p.L947S) displayed a more minor effect. However, for these 2 VUS, we found that the intensity of RAD51 foci was dramatically reduced (Fig. 7e). As all 5 variants displayed problems in protein stability in mES cells, we believe that the defects observed in RAD51 foci formation and/or intensity mostly stem from impaired RAD51 recruitment due to reduced PALB2 protein levels. Overall, our findings in human cell-based assays solidify those obtained in the mES cell-based assays, indicating that our system in mES cells is robust and suited for semi-high throughput functional analysis of VUS in human *PALB2*.

## Discussion

To address the impact of *PALB2* VUS on protein function, we developed a mES cell-based system that allows a rapid and robust functional classification of genetic variants in human *PALB2*. Out

of the 49 PALB2 missense variants tested in this study (Supplementary Data 1), we identified 15 variants (p.L24S, p.Y28C, p.L35P, p.W912G, p.G937R, p.I944N, p.L947S, p.L961P, p.L972Q, p.A1025R, p.T1030I, p.I1037T, p.G1043D, p.L1070P, and p.L1172P) as damaging, reducing HR by >60%. For three variants that have been described previously (p.Y28C, p.L35P, and p.T1030I), our results are highly consistent with published data, showing that these variants which confer increased risk for breast cancer, strongly impact HR[30,31]. Furthermore, we observed a strong positive correlation between the DR-GFP and PARPi or cisplatin sensitivity assays, suggesting that carriers of the identified damaging VUS may benefit from PARPi- or cisplatin-based treatment. Lastly, our data from the human cell-based assays further verify the results from the mES cell-based assays, indicating that our system in mES cells is well-suited for the rapid, semi-high throughput functional analysis of VUS in human *PALB2*.

In addition to p.Y28C and p.L35P, which have both been reported to impair the interaction with BRCA1[30], p.R37H also resides in the N-terminal CC domain and impairs the HR activity by more than 55% in our DR-GFP assay (Fig. 2b). In contrast to an earlier report showing that p.R37H did not affect the interaction with BRCA1[30], we found that this variant impaired the PALB2-BRCA1 interaction and the BRCA1-dependent

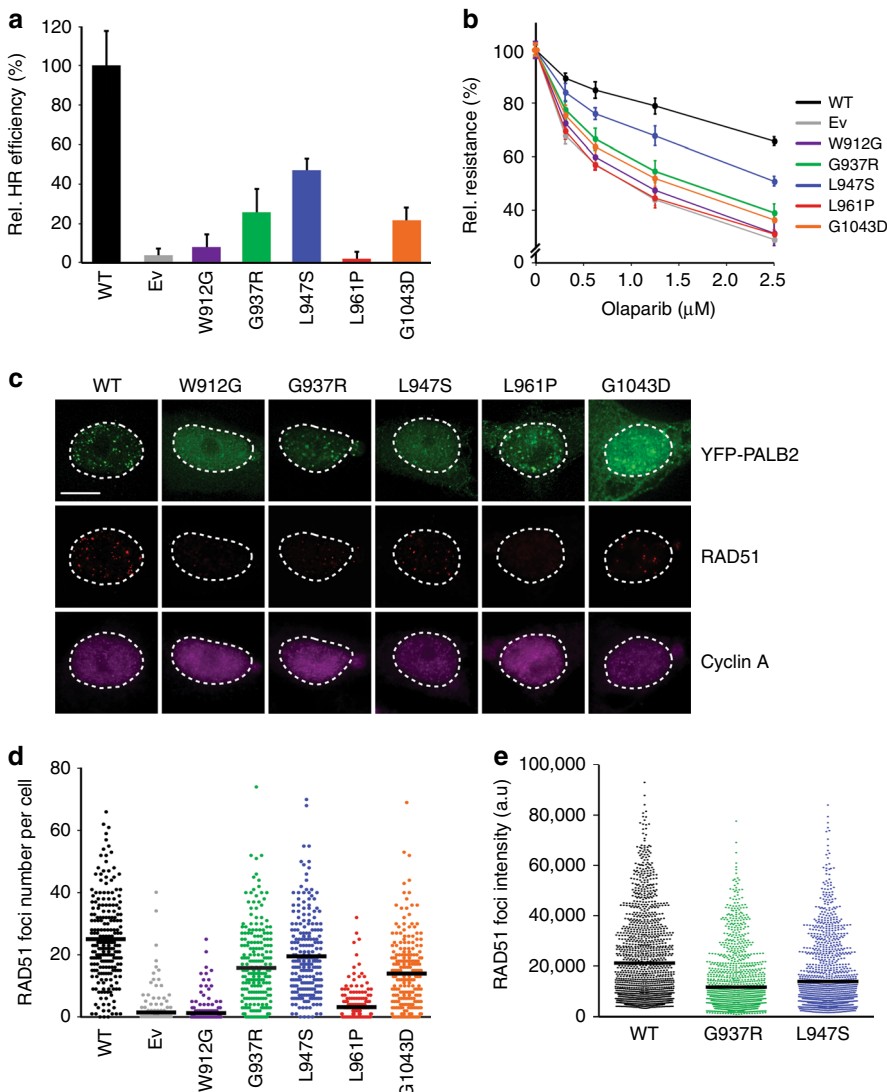

**Fig. 7** Functional analysis of damaging *PALB2* variants in human cells. **a** CRISPR-LMNA HDR assay in siRNA-treated U2OS *PALB2* knockdown cells expressing siRNA-resistant human *PALB2* cDNA with the indicated variants (or an empty vector control, Ev). Data represent the mean percentage (±SD) of mRuby2-positive cells among the YFP-positive cells from three independent experiments (*n* > 300 YFP-positive cells per condition) relative to wild type (WT), which was set to 100%. **b** PARP inhibitor (PARPi) sensitivity assay using siRNA-treated HeLa *PALB2* knockdown cells expressing siRNA-resistant human *PALB2* cDNA with the indicated variants (or an empty vector control, Ev). Survival curves were determined after 72 h of PARPi treatment. Data represent the mean percentage of viability relative to untreated cells ( ± SD), which was set to 100%, of three independent experiments, each performed in triplicate. **c** Representative images of RAD51 foci 4 h after 2 Gy of ionizing radiation in siRNA-treated HeLa *PALB2* knockdown cells expressing siRNA-resistant human *PALB2* cDNA with the indicated variants (or an empty vector control, Ev). Scale bar: 5 μm. **d** Quantification of the results from (**c**). Scatter dot plot shows the number of RAD51 foci in cyclin A-positive cells expressing the indicated variant, with the horizontal lines designating the mean values (±SD) of three independent experiments (*n* > 200 cells per condition). **e** Quantification of the results from (**c**). Scatter dot plot shows the intensity of RAD51 foci in cyclin A-positive cells expressing the indicated variant, with the horizontal lines designating the mean values (±SD) of three independent experiments (*n* > 500 cells per condition). Source data are provided as a Source Data file.

recruitment of PALB2 to sites of DNA damage, which is highly consistent with its moderate impact on HR. Our results on the identified p.L24S variant, are in line with a previous study in which the CC6 PALB2 variant, for which the amino acids LKK at position 24–26 are changed to AAA, impairs the interaction with BRCA1 and consequently abrogates HR[42]. Thus, our HR and protein-protein association studies for both p.L24S and p.L35P further underline the importance of the BRCA1-PALB2 interaction for efficient HR and likely tumor suppression.

The C-terminal WD40 domain of PALB2 is an important regulatory platform that mediates interactions with several important HR pathway components, such as BRCA2 and RAD51. Crystal structure studies of the WD40 domain showed that it

forms a seven-bladed β-propeller-like structure of which correct folding is crucial for PALB2 function[29]. As such, it is likely that variants in this region are prone to interfere with the structure and/or biochemical properties of this domain. For example, although it has been reported that p.W1038X exposes a nuclear export signal leading to cytoplasmic localization[43], we see in our assays that the expression levels of this variant are dramatically reduced compared to wild-type PALB2 (Fig. 4a), probably due to instability/misfolding and rapid degradation in the cytoplasm. Indeed, we see similar effects for three other truncating variants (p.Q899X, p.P1009Lfs, and p.Y1183X). Consistent with the WD40 domain being prone to 'destabilizing' variants, we identified 11 damaging VUS in the WD40 domain that exhibited

strongly reduced PALB2 protein levels, and consequently strongly reduced HR (~60–95%). Importantly, 5 of these 11 VUS are bona fide null variants that abrogate the HR activity to the same extent as the *PALB2* truncating variants. These results indicate that the WD40 domain is a 'hotspot' for deleterious variants that affect protein stability. Consistently, a recent study on PTEN, showed that 64% of the pathogenic missense variants reduce its expression level[44]. This suggests that protein instability due to LOF variants in tumor suppressor genes, including *PALB2*, constitutes a mechanism of pathogenicity.

Several studies have implicated *BRCA1*, *BRCA2*, and *PALB2* in DNA-damage-induced checkpoint control[38–40]. Accordingly, we found that G2/M checkpoint maintenance after IR is compromised in *Trp53^KO^/Palb2^KO^* mES cells, an effect that could be rescued by expressing wild-type human *PALB2*. Interestingly, *PALB2* variants that show LOF in HR, were unable to maintain an efficient G2/M checkpoint response. Both p.L35P and p. A1025R, which are unable to interact with BRCA1 and BRCA2, respectively, were among these variants, suggesting that these interactions are key to PALB2's checkpoint function. Moreover, we infer that the observed defects in G2/M checkpoint maintenance could stem from defective HR. In line with such a scenario, an inverse correlation has been observed between HR activity and POLQ-mediated DSB repair[45]. This indicates that POLQ-mediated DSB repair may act as a compensatory pathway for PALB2-dependent HR that potentially affects G2/M checkpoint maintenance in response to DNA breaks.

Although our functional assays may aid in the classification of rare *PALB2* VUS, a major challenge will be to translate effects on PALB2 protein function into estimates for cancer risk. Whereas the truncating *PALB2* variants have been associated with an odds ratio of 7.46[5], the p.L939W variant has been associated with an odds ratio of 1.05[46]. This would suggest that a decrease of 40% in HR in our DR-GFP assay, as shown for the p.L939W variant (Fig. 2b), would barely increase the risk for breast cancer. Therefore, it will be interesting to see whether the extent to which variants affect HR is proportional to increased cancer risk and at which level of HR deficiency, cancer risk significantly increases. Finally, it will be important to examine whether *PALB2* VUS, either in coding or non-coding sequences, affect *PALB2* splicing. For all missense variants presented in this study in silico splice site prediction analysis was performed using five different algorithms (Splice Site Finder-like, MaxEntScan, GeneSplicer, NNSplice, Human Splicing Finder) in Alamut (http://www.interactive-biosoftware.com/). For all VUS, an effect on RNA splicing was unlikely, with the exception of c.53A>G (p.K18R) for which NNSplice predicted the introduction of a new weak acceptor recognition site in exon 2. Complementation of our *Trp53^KO^/Palb2^KO^* cells with a bacterial artificial chromosome (BAC) containing the full length human *PALB2* gene, as has been previously described for *BRCA1* and *BRCA2*[14,47,48], would allow addressing the functional effect of splice variants in *PALB2*. Ultimately, the results from functional assays for VUS can be incorporated into multifactorial risk models to allow for better clinical classification in the future. Indeed, multiple pieces of evidence, in addition to functional assay results, will be required to enable the clinical classification of VUS.

## Methods

**Cell lines and culture conditions**. 129/Ola E14 IB10 mES cells[49] were cultured on gelatin-coated dishes in 50% BRL/50% complete medium[13] with 0.1 mM beta-mercaptoethanol (Merck) and 10³ U/ml ESGRO LIF (Millipore). STR genotyped U2OS and HeLa human cells (ATCC) were maintained, respectively, in McCoy's 5A (Wisent) and DMEM (ThermoFischer) supplemented with 10% Fetal Bovine Serum (FBS) and 1% penicillin and streptomycin.

**Generation of *Trp53^KO^/Palb2^KO^* mES cells with DR-GFP and RMCE**. *Trp53^KO^/Palb2^KO^* mES cells carrying the DR-GFP reporter and RMCE system were generated as follows. 75 μg of the plasmids carrying *Pim1*:DR-GFP (p59X DRGFP)[50] or the *Rosa26*:RMCE acceptor cassette (pTT5-Puro) (TaconicArtemis GmbH) were linearized with *Xho*I and *Pvu*I respectively. *Pim1*:DR-GFP was transfected into mES cells[49] using Lipofectamine 2000 (Invitrogen). Integration of DR-GFP at *Pim1* was verified using PCR and Southern blot analysis. Similarly, the RMCE acceptor cassette was integrated at *Rosa26* in cells carrying DR-GFP. Integration of the RMCE acceptor cassette at *Rosa26* was verified using PCR and Southern blot analysis. *Trp53^KO^* cells were generated by transfection of 1 μg of pSpCas9(BB)-2A-GFP (pX458)[51], which encodes a gRNA that targets exon 1 (5'-CGAGCTCCCTCTGAGCCAGG-3'), into mES cells carrying DR-GFP and the RMCE acceptor cassette. GFP-positive cells were FACS-sorted and seeded. Individual clones were examined by TIDE and western blot analysis for loss of p53 expression. Similarly, the *Palb2^KO^* was generated in *Trp53^KO^* mES cells carrying DR-GFP and RMCE acceptor cassette using a gRNA that targets exon 4 (5'-GGGGACAACAAAGACGCCGT-3'), and verified by TIDE and western blot analysis for loss of Palb2 expression.

**Cloning and site-directed mutagenesis of human *PALB2* cDNA**. pBudCE4.1 (ThermoFisher, V53220), which contains an EF1α promotor, was modified by cloning two different oligonucleotides with *Pac*I restriction sites into the *Nhe*I (5'-CTAGGACTTAATTAAGTCGATCGCCGG-3') and *Bgl*II restriction sites (5'-GATCTCTTAATTAAGACTG-3'), respectively. Human FLAG-tagged *PALB2* cDNA was obtained from pcDNA3-FLAG-PALB2 and subcloned into pBudCE4.1-*Pac*I using the *Acc*65I and *Xho*I restriction sites. An Ef1α-*PALB2*-containing fragment from pBudCE4.1-*Pac*I-PALB2 was then cloned into the RMCE vector pRNA-251-MCS-RMCE (TaconicArtemis GmbH) using the *Pac*I restriction sites in both vectors. *PALB2* variants were introduced by site-directed mutagenesis using the Quick-Change Lightning protocol (Agilent Technologies). Constructs were verified by Sanger sequencing and used for mES cell-based assays. For human cell-based assays, siRNA-resistant pEYFP-PALB2 construct was generated by site-directed mutagenesis using the Q5 Site-Directed Mutagenesis Kit (New England Biolabs) as per the manufacturer's protocol with the following primers: forward primer—5'-GATCTTATTGTTCTACCAGGAAAATC-3' and reverse primer—5'-TTCCTCTAAGTCCTCCATTTCTG-3'. PALB2 variants were introduced in the siRNA-resistant pEYFP-C1-PALB2 plasmid by site-directed mutagenesis using the same kit. All primers used for site-directed mutagenesis are listed in Supplementary Data 1.

**Karyotyping**. mES cells (50% confluency) were incubated with 0.05 μg/ml colcemid (Gibco) for 2.5 h. After trypsinization, 2.5 ml of 0.4% Na-citrate/0.4% KCL (1:1) was added in a dropwise manner. Cells were centrifuged at 120 × *g* after which the supernatant was aspirated and 2.5 ml fixative consisting of methanol and acidic acid (4:1) was added while slowly vortexing. This step was repeated twice. Using ultrathin pipet tips, a small number of cells was dropped onto a cleaned microscopy slide (VWR, 631–1551) and left to air-dry. DAPI was used for visualizing the chromosomes, which were counted using a Zeiss microscope Imager M2 (×63) and ZEN 2012 microscopy software.

**Western blot analysis**. Expression of endogenous mouse PALB2 and human PALB2 in mES was monitored by protein extraction and western blot. Briefly, protein samples were generated by taking up ~1.5 × 10⁶ cells in 75 μl Laemmli buffer and boiling them at 95 °C for 5 min. Samples were incubated with 1.5 μl benzonase (Merck Millipore 70746-3, 25 U/μl) for 10 min at room temperature and then loaded for gel electrophoresis and immunoblotting. Primary antibodies used were a rabbit polyclonal antibody against the N-terminus of human PALB2 (1:1000, kindly provided by Cell Signaling Technology prior to commercialization), a homemade rabbit antibody against the N-terminus of mouse PALB2[42] (NB3 anti-mPalb2, 1:2000, kind gift from Bing Xia) and a mouse monoclonal antibody against alpha tubulin (1:10,000, Sigma, T6199 clone DM1A). Peroxidase-AffiniPure Goat Anti-Rabbit secondary antibody (Jackson laboratories) and SuperSignal West Femto Maximum Sensitivity Substrate (ThermoFisher) were used for development of blots on the Amersham Imager 600 (GE Healthcare Life Sciences).

Western blotting was performed by separating U2OS and HeLa protein extracts on 12% SDS-PAGE gels at 100 V and transferred to nitrocellulose membrane during 1.5 h at 100 V. Membranes were blocked for 1 h in 5% milk in Tris-buffered saline (TBS)-Tween. Primary antibodies applied were mouse monoclonal anti-GFP (1:1000, Roche, #11814460001), anti-alpha tubulin (1:200000, Abcam, #ab7291) and a home-made rabbit polyclonal antibody against human PALB2 (1:5000). Horseradish peroxidase-conjugated anti-mouse IgG (1:10000, Jackson ImmunoResearch) was used as secondary antibody.

**RT-qPCR analysis**. RNA was isolated from mES cells on 6-well plates using Trizol (ThermoFisher, 15596026) as per the manufacturer's protocol. For each condition, 3 μg RNA was treated with RQ1 RNAse-free DNAse (Promega, M6101) and cDNA was synthesized from 0.2 μg DNAse-treated RNA using hexamer primers (ThermoFisher, N8080127) and AMV Reverse Transcriptase (ThermoFisher, 12328019) as per the manufacturer's protocols. RT-qPCRs were carried out using GoTaq

qPCR Master mix (Promega, A6002), a CFX384 Real-Time System (Bio-Rad) and the following qPCR primers directed at the human *PALB2* cDNA or the mouse control gene *Pim1*; human PALB2-Fw— 5′-GATTACAAGGATGACGACGATA AGATGGAC-3′, human PALB2-Rv—5′-CCTTTTCAAGAATGCTAATTTCTCC TTTAACTTTTCC-3′, mouse Pim1-exon4-Fw—5′-GCGGCGAAATCAAACTCA TCGAC-3′, and mouse Pim1-exon5-Rv—5′-GTAGCGATGGTAGCGAATCCAC TCTGG-3′.

**HR reporter assays**. $2 \times 10^6$ *Trp53*[KO]/*Palb2*[KO] mES cells carrying the DR-GFP reporter and RMCE system were subjected to RMCE by co-transfecting 1 μg FlpO expression vector (pCAGGs-FlpO-IRES-puro)[19] with 1 μg RMCE exchange vector. Neomycin-resistant cells from ~500 resistant clones were pooled and expanded prior to use in DR-GFP reporter assays. $1 \times 10^6$ cells were transfected with 1 μg of plasmid for co-expression of I-SceI and mCherry (pCMV-Red-Isce, kind gift from Jos Jonkers) using Lipofectamine 2000 (ThermoFisher)[13]. A co-transfection of 1 μg pCAGGs[52] with 0.05 μg of an mCherry expression vector was included as control. Two days after transfection, mCherry/GFP double-positive cells were scored using a Novocyte Flow Cytometer (ACEA Biosciences, Inc.).

For the CRISPR-LMNA HR assay[43], U2OS cells were seeded in 6-well plates at $2 \times 10^5$ cells per well. Knockdown of *PALB2* was performed 6 h later with 50 nM siRNA against PALB2 (5′-CUUAGAAGAGGACCUUAUU-3′; Dharmacon) using Lipofectamine RNAiMAX (Invitrogen). 24 h post-transfection, $1.5 \times 10^6$ cells were pelleted for each condition and resuspended in 100 μl complete nucleofector solution (SE Cell Line 4D-Nucleofector™ X Kit, Lonza) to which 1 μg of pCR2.1-mRuby2LMNAdonor, 1 μg pX330-LMNAgRNA, 1 μg pEYFP-C1 or the indicated siRNA-resistant YFP-PALB2 construct, and 150 pmol siRNA was added. Once transferred to a 100 μl Lonza certified cuvette, cells were transfected using the 4D-Nucleofector X-unit, program CM-104 and transferred to a 10 cm dish. After 48 h, cells were trypsinized and plated onto glass coverslips. Cells were fixed with 4% paraformaldehyde and analyzed for mRuby2 and YFP expression on a Leica CTR 6000 inverted microscope using a 63×/1.40 oil immersion objective 72 h post-nucleofection.

**PARPi and cisplatin sensitivity assays**. For proliferation-based PARPi and cisplatin sensitivity assays, mES cells were seeded in triplicate at 10.000 cells per well of a 96-well plate. The next day, cells were treated with PARP inhibitor Olaparib (Selleckchem, S1060) or cisplatin (Accord Healthcare, 15683354) for two days, after which the medium was refreshed and cells were cultured for one more day. Viable cells were subsequently counted using the Novocyte Flow Cytometer (ACEA Biosciences, Inc.).

For clonogenic PARPi survival assays, mES cells were seeded on p60 plates at the following densities: 250 cells without PARPi, 400 cells for functional variants with 1 or 5 nM PARPi, and 3000 cells for damaging variants with 1 or 5 nM PARPi. Cells were treated for 7–9 days allowing the visible formation of surviving colonies, which were counted following methylene blue staining (2.5 gr/L in 5% ethanol). HeLa cells were seeded at 240,000 cells per well of a 6-well plate before being transfected 6 h later with 50 nM control or *PALB2* siRNA using Lipofectamine RNAiMAX (Invitrogen). The next day, cells were complemented with 0.8 μg of pEYFP-PALB2 plasmid DNA using Lipofectamine 2000 (Invitrogen) for 24 h and then seeded in triplicates into a Corning 3603 black-sided clear bottom 96-well microplate at a density of 3000 cells per well. After 3 days of treatment with Olaparib (Selleckchem, S1060), nuclei were stained with Hoechst 33342 (Invitrogen) at 10 μg/ml in media for 45 min at 37 °C. Images of entire wells were acquired at 4x magnification with a Cytation 5 Cell Imaging Multi-Mode Reader followed by quantification of Hoechst-stained nuclei with the Gen5 Data Analysis Software v3.03 (BioTek Instruments).

**Cell cycle analysis and G2/M checkpoint assays**. For cell cycle profile analysis cells were fixed in 70% ethanol. After 15 min incubation on ice, cells were pelleted and resuspended in 500 μl PBS containing 50 μg/ml propidium iodide (PI) (ThermoFisher, P1304MP), 0.1 mg/ml RNase A and 0.05% Triton X-100, followed by 40 min incubation at 37 °C. Cells were then washed with PBS and analyzed using the Novocyte Flow Cytometer (ACEA Biosciences, Inc.).

For G2/M checkpoint assays, $1 \times 10^6$ mES cells were seeded on p60 dishes one day before exposure to 3 or 10 Gy of IR. One or 6 h later, cells were fixed as described for cell cycle profile analysis and incubated overnight at −20 °C. Fixed cells were then permeabilized for 15 min on ice using 0.25% Triton X-100 in PBS, after which mitotic cells were stained in 100 μl PBS with 1 μl anti-phospho-H3 Ser10 (1 μg/μl, Sigma-Aldrich, 06–570) for 3 h at room temperature. Alexa-488 goat α-rabbit (1:100 in 100 μl PBS; ThermoFisher, 11034) was used as a secondary antibody. Cells were analyzed using the Novocyte Flow Cytometer (ACEA Biosciences, Inc.).

**Pulldown assays**. 20 μg pYFP-PALB2 plasmid [53] was transfected into ~$10 \times 10^6$ U2OS cells on a 15 cm dish using Lipofectamine 2000. The next day cells were trypsinized, washed with cold PBS, and transferred to LoBind Eppendorf tubes. Cells were then lysed in 1 ml EBC buffer (50 mM Tris pH 7.3, 150 mM NaCl, 0.5% NP-40, 2.5 mM $MgCl_2$), containing 1 tablet protease inhibitor (Roche) per 10 ml buffer. 500 Units benzonase was then added to each condition and cells were

incubated for 60 min at 4 °C on a rotating wheel. The lysate was subsequently centrifuged for 10 min at $18,400 \times g$ at 4 °C. The supernatant was then added to 25 μl of pre-washed GFP-trap beads (ChromoTek) in LoBind Eppendorf tubes and incubated for 1.5 h at 4 °C on a rotating wheel. The beads were washed 5–6 times with EBC buffer with spinning steps of 1 min at 3380 g at 4 °C. Beads were eventually resuspended in 25 μl Laemmli buffer after which about half of each sample was analyzed by western blot analysis using a homemade rabbit antibody against human BRCA1[54] (1:1000, kind gift form Dan Durocher).

**Laser micro-irradiation and PALB2 recruitment**. U2OS cells were grown on 18-mm coverslips and sensitized with 10 μM 5′-bromo-2-deoxyuridine (BrdU) for 24 h before micro-irradiation. Cells were co-transfected with 1 μg pYFP-PALB2, with or without a variant, and 0.5 μg mCherry-NBS1 expression vector using lipofectamine 2000 (Invitrogen). For micro-irradiation, cells were placed in a live-cell imaging chamber set to 37 °C in $CO_2$-independent Leibovitz's L15 medium supplemented with 10% FCS and penicillin–streptomycin (Invitrogen). Live cell imaging and micro-irradiation experiments were carried out with a Zeiss Axio Observer microscope driven by ZEN software using a ×63/1.4 oil immersion objective coupled to a 355 nm pulsed DPSS UV-laser (Rapp OptoElectronic). To monitor the recruitment of YFP-PALB2 to laser-induced DNA damage sites, cells were imaged before and after laser irradiation at 90 s time intervals over a period of 10.5 min. The fluorescence intensity of YFP-PALB2 and mCherry-NBS1 at DNA damage sites relative to that in an unirradiated region of the nucleus was quantified and plotted over time. Kinetic curves were obtained by averaging the relative fluorescence intensity of cells displaying positive recruitment ($n > 30$ cells per condition).

**RAD51 foci analysis**. HeLa cells were seeded on glass coverslips in 6-well plates at 225.000 cells per well. Knockdown of *PALB2* was performed 18 h later with 50 nM PALB2 siRNA using Lipofectamine RNAiMAX (Invitrogen). After 5 h, cells were subjected to a double thymidine block. Briefly, cells were treated with 2 mM thymidine for 18 h and released into fresh medium for 9 h. During the release time, 0.8 μg pEYFP-PALB2 plasmid DNA (with or without variant) was transfected into the cells using Lipofectamine 2000 (Invitrogen). Cells were then treated with 2 mM thymidine for 17 h and protected from light from this point on. After 2 h of release from the second block, cells were irradiated with 2 Gy and processed for immunofluorescence 4 h post-irradiation. Unless otherwise stated, all immunofluorescence dilutions were prepared in PBS and incubations performed at room temperature with intervening washes in PBS. Cell fixation was carried out by incubation with 4% paraformaldehyde for 10 min followed by 100% ice-cold methanol for 5 min at −20 °C. This was succeeded by permeabilization in 0.2% Triton X-100 for 5 min and a quenching step using 0.1% sodium borohydride for 5 min. After blocking for 1 h in a solution containing 10% goat serum and 1% BSA, cells were incubated for 1 h with primary antibodies anti-RAD51 (1 :7000, B-bridge International, #70-001) and anti-cyclin A (1 :400, BD Biosciences, # 611268) diluted in 1% BSA. Secondary antibodies Alexa Fluor 568 goat anti-rabbit (Invitrogen, #A-11011) and Alexa Fluor 647 goat anti-mouse (Invitrogen, #A-21235) were diluted 1 :1000 in 1% BSA and applied for 1 h. Nuclei were stained for 10 min with 1 μg/mL DAPI prior to mounting onto slides with 90% glycerol containing 1 mg/ml paraphenylenediamine anti-fade reagent. Z-stack images were acquired at ×63 magnification on a Leica CTR 6000 microscope, then deconvolved and analyzed for RAD51 foci. The number and intensity of RAD51 foci in cyclin A-positive cells expressing the indicated pEYFP-PALB2 constructs were scored using automatic spot counting in Velocity software v6.0.1 (Perkin-Elmer Improvision).

**Reporting summary**. Further information on research design is available in the Nature Research Reporting Summary linked to this article.

## Data availability

All data generated or analyzed during this study are included in this published article (and its Supplementary information files).

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

## Acknowledgements

The authors would like to thank Jos Jonkers and Peter Bouwman for providing the pTT5-Puro (RMCE acceptor cassette), pRNA-251-MCS-RMCE (RMCE exchange vector) and pCMV-Red-I-SceI constructs, as well as Dan Durocher and Bing Xia for sharing BRCA1 and PALB2 antibodies, respectively. We are grateful to Maria Jasin and Francis Stewart for sharing the DR-GFP reporter and FlpO constructs, respectively. We thank Patrick van Vliet and Richard Lemmers for help with the Southern blot analysis, and Cell Signaling for providing antibodies directed against PALB2 prior to commercialization. Finally, we would like to thank Jamie Allen and the BRIDGES consortium for providing *PALB2* variants. This work was financially supported by the Government of Canada through Genome Canada and the Canadian Institutes of Health Research, the Ministère de l'Économie, de la Science et de l'Innovation du Québec through Genome Québec and the Quebec Breast Cancer Foundation (J.S. and J.-Y.M.), as well as by grants from the Ministère de l'Économie, de la Science et et l'Innovation du Québec through the PSR-SIIRI-949 program (J.S. and J.-Y.M), the CIHR (Foundation grant to J.-Y.M), European Union (BRIDGES grant to P.D., M.V., and H.v.A.) and the Dutch Cancer Society (P.D. and H.v.A.).

## Author contributions

R.B. cloned *PALB2* cDNA in the RMCE exchange construct, generated *PALB2* variants using site-directed mutagenesis, generated *Trp53^{KO}/Palb2^{KO}* mES cells harboring the DR-GFP reporter and RMCE acceptor cassette, and performed DR-GFP, PARPi,

Cisplatin, G2/M checkpoint, and pulldown assays, as well as Southern and western blot analysis, and PCR and DNA sequencing analysis in mES cells. Amélie Rodrigue generated YFP-PALB2 variants using site-directed mutagenesis and performed CRISPR-LMNA HR, PARPi sensitivity and RAD51 foci assays, as well as western blot analysis in human cells. C.S. assisted with the PARPi assays in mES cells. W.W. performed G2/M checkpoint assays. B.V. performed in silico modeling of *PALB2* variants. M.S. generated PALB2 p.A1025R using site-directed mutagenesis and performed DR-GFP and PARPi proliferation assays for this variant. M.R. studied YFP-PALB2 recruitment to laser-induced DNA damage. N.C. performed cell cycle profile analysis in mES cells. M.V. helped with gathering *PALB2* variants from databases and literature and assisted in the in silico analysis. F.C. provided p.L24S, p.I944N and p.L1070P variants. J.S. and J.-Y.M. supervised experiments performed in human cells. M.V., P.D., and H.v.A. conceived the project. H.v.A. supervised the project. R.B and H.v.A. wrote the paper.

## Competing interests
The authors declare no competing interests.

## Additional information

**Supplementary information** is avaliable for this paper at https://doi.org/10.1038/s41467-019-13194-2.

