## [Peer Review File · Nature Communications]

Reviewers' comments:

Reviewer #1 (Remarks to the Author):

The manuscript by Boonen et al analyzes VUS in PALB2 for HR proficiency and ultimately predictive response to therapeutic strategies for breast cancer patients. The authors generate a mouse cell model that mimics the tumor setting in which p53 deficiency allows for loss of functional PALB2. This highly relevant work establishes a semi-high throughput method of testing VUS that lack functional analysis and could be applied to all VUS in PALB2 in the future for diagnostic purposes. The authors further establish mechanistic insight to domains of PALB2 through the clustering of VUS phenotypes. They establish that functional testing is more accurate than predictive analysis by online software or in silico modeling. One of the limitations of this manuscript is the lack of new functional insight into PALB2. Although these results will be of interest to both clinical and basic science audiences, this knowledge doesn't uncover new mechanisms into PALB2 function in tumorigenesis.

Major Comments:

1. PALB2 is important for both canonical HR and ICL repair. While the DR-GFP assay and PARPi sensitivity analyze proficiency for repairing a direct DSB by HR, the VUS may have separate phenotypes for ICL repair. To address this gap, the VUS that were proficient for the DR-GFP assay should be tested for MMC sensitivity or another readout of ICL repair. These results may indicate patients with these VUS who will benefit from an alternative therapeutic strategy.

Minor Comments:

1. While PARPi proliferation sensitivity is a semi-high throughput analysis for VUS function, will this correlate to longer-term sensitivity of VUS to PARPi. It would be informative to select some PALB2 HR-proficient and some HR-deficient mutants for PARPi sensitivity by clonogenic survival. Perhaps the 4 novel VUS (W912G, L961P, I944N, G1043D) would be of most interest to readers.

Reviewer #2 (Remarks to the Author):

This interesting paper reports the development of an assay system to evaluate the function of mis-sense mutations in the cancer suppressor PALB2, and its use to show that certain PALB2 variants of previously uncertain pathogenic significance (VUSs) affect aspects of biochemical or cellular function.

To develop their assay system, the authors integrate the well-established DR-GFP reporter for measuring proficiency in homologous recombination (HR) into the genome of murine embryonic stem (mES) cells, then delete endogenous murine p53 and Palb2 genes, while introducing various forms of human PALB2 cDNA into the Rosa26 locus. The functional effect of 48 different PALB2 VUSs on HR was then examined, leading to the determination that several different VUSs are potentially deleterious to PALB2 function. Further functional analyses were carried out to validate that these VUSs affect known PALB2 functions relevant to HR.

The strengths of the paper are that it is (generally) technically sound and well written. The major weakness is that the work represents a limited scientific and technical advance, which is more suited to a specialist journal. In addition, several factors limit the potential significance and impact of the work.

Conceptually similar assays to determine the functional effects of VUSs have already been reported for related proteins like BRCA1 and BRCA2 (eg., Chang et al J Clin Invest 2009; Biswas et

al Hum Mol Genet 2012; Guidugli et al Hum Mutat 2014; Guidugli et al Am J Hum Genet 2018). In addition, more recent work reports conceptually novel approaches using saturation genome editing to address the assignment of VUSs in BRCA1 (eg., Findlay et al Nature 2018). Placed in the context of this prior literature, the manuscript represents a somewhat limited scientific and technical advance.

PALB2 has been implicated in cellular functions other than HR (eg., Simhadri et al., Oncogene 2019; Ma et al MCB 2012) that are plausibly relevant to cancer suppression. It remains possible that VUSs that do not score in the HR assays reported in this paper might nevertheless be deleterious.

Finally, characterization of the potential mechanism of VUSs identified here remains preliminary. For example, p.R37H in the PALB2 N-terminus impairs HR but apparently does not affect the BRCA1-PALB2 interaction. Much further work is necessary to establish how this mutation may work to affect PALB2 function.

Reviewer #3 (Remarks to the Author):

The authors describe an elegant in vivo assay that allows the effects of single-nucleotide variants on the function of the HR scaffold protein PALB2 to be examined.

They describe several sets of experiments which look at Variants of Unknown Significance (VUS), their effects on the homologous recombination pathway of DNA repair and the sensitisation of cells to PARP inhibition. They also go onto validate their results for selected VUS in human cells.

Overall the manuscript is of a high standard, and should be considered for publication after some minor revision.

MAJOR POINTS:

> Panel C of Figure 4 is very difficult to interpret or follow — and actually only shows the environment of the native amino acid, and not the predicted effect of the SNP / VUS — for a reader well-versed in structural biology this does not represent a particular issue, but a more general audience would find this problematic. This figure should be revisited and revised. A key should also be provided alongside the figure to aid the reader.

> The manuscript also describes each the predicted structural affects of each VUS in a different order to that presented in the figure, again potentially affecting the manuscript's readability and ease of interpretation.

Page 12: However, it is important to note that in the set of PALB2 missense variants analyzed in this study, there are several variant for which similar destabilising effects would be predicted by using in silico modeling, but which we found to be (completely) functional in our HR-assays

> Which missense variants ARE these? This currently reads a little like a throw-away unsupported statement.

Page 6; semi high-throughput

> Can an assay be described as 'semi' high-throughput? Consider changing to medium throughput.

Page 6; HR is highly efficient in this phase of the cell cycle

>HR in S-phase is actually reasonably restricted, as HR cannot occur until a sister chromatid template becomes available — which itself is determined by when a particular stretch of DNA is replicated, and hence to whether it is in proximity to an early or late-firing origin. Please consider rephrasing accordingly.

Point-by-point response to the reviewer's comments

NCOMMS-19-10676

Functional analysis of genetic variants in the high-risk breast cancer susceptibility gene *PALB2*

Rick A.C.M. Boonen, Amélie Rodrigue, Chantal Stoepker, **Wouter W. Wiegant**, Bas Vroling, **Milan Sharma**, **Magda Rother**, Nandi Celosse, Maaïke P.G. Vreeswijk, Fergus Couch, Jacques Simard, Peter Devilee, Jean-Yves Masson and Haico van Attikum

We would like to thank the reviewers for their positive feedback and constructive comments on our manuscript. Based on this we have performed several additional experiments, which were added to the manuscript. The results of these experiments fully support and extend the conclusions of our work. We have also added two new supplementary figures showing the original uncropped DNA gels and western blots. Below you will find a point-by-point response to the comments (text in red). All changes to the manuscript are also indicated with text in red.

Reviewer #1 (Remarks to the Author):

The manuscript by Boonen et al analyzes VUS in PALB2 for HR proficiency and ultimately predictive response to therapeutic strategies for breast cancer patients. The authors generate a mouse cell model that mimics the tumor setting in which p53 deficiency allows for loss of functional PALB2. This highly relevant work establishes a semi-high throughput method of testing VUS that lack functional analysis and could be applied to all VUS in PALB2 in the future for diagnostic purposes. The authors further establish mechanistic insight to domains of PALB2 through the clustering of VUS phenotypes. They establish that functional testing is more accurate than predictive analysis by online software or in silico modeling. One of the limitations of this manuscript is the lack of new functional insight into PALB2. Although these results will be of interest to both clinical and basic science audiences, this knowledge doesn't uncover new mechanisms into PALB2 function in tumorigenesis.

Major Comments:

1. PALB2 is important for both canonical HR and ICL repair. While the DR-GFP assay and PARPi sensitivity analyze proficiency for repairing a direct DSB by HR, the VUS may have separate phenotypes for ICL repair. To address this gap, the VUS that were proficient for the DR-GFP assay should be tested for MMC sensitivity or another readout of ICL repair. These results may indicate patients with these VUS who will benefit from an alternative therapeutic strategy.

We have arbitrarily selected 11 VUS (Y1064C, S1058P, L931R, E1018D, P707L, Y408H, E42K, D871G, K18R L939W, R37H) that did not grossly affect HR in DR-GFP and PARPi sensitivity assays (Figs. 2 and 3A). As negative controls two benign variants (D134N, G998E) that were HR-proficient were included, while two truncating (Y551X, Y1183X) and three VUS (G1043D, L961P, L35P) that were HR-deficient served as positive controls (Figs. 2 and 3A). Subsequently, we tested whether PALB2 knockout (KO) cells expressing these variants

display sensitivity to cisplatin, a readout that served as a measure for the repair efficiency of ICLs induced by this DNA damaging agent. Strikingly, the 11 VUS that did not grossly impact HR, as well as the 2 benign variants, did not confer sensitivity to cisplatin treatment when compared to that of cells expressing wild-type (WT) PALB2 (**see new results in Fig. 3d**). In contrast, the 3 VUS and 2 truncating variants that impaired HR, rendered cells hypersensitive to cisplatin treatment comparable to that observed for PALB2 KO cells expressing empty vector (Ev) (**see new results in Fig. 3d**). Importantly, when comparing the HR efficiency measured by DR-GFP and cisplatin sensitivity assays, a strong positive correlation was observed for all variants tested ($R^2=0.8313$) (**see new results in Fig. 3e**). This is consistent with the fact that DNA double-strand breaks that arise during ICL repair in S-phase cells are repaired by PALB2-dependent HR (1). These results indicate that patients with VUS in PALB2 that do not impair HR (as measured in DR-GFP and PARPi sensitivity assays) may not necessarily benefit from an alternative therapeutic strategy based on chemotherapy with ICL-inducing agents.

Minor Comments:

1. While PARPi proliferation sensitivity is a semi-high throughput analysis for VUS function, will this correlate to longer-term sensitivity of VUS to PARPi. It would be informative to select some PALB2 HR-proficient and some HR-deficient mutants for PARPi sensitivity by clonogenic survival. Perhaps the 4 novel VUS (W912G, L961P, I944N, G1043D) would be of most interest to readers.

We have tested whether PALB2 KO cells expressing these 4 novel VUS (W912G, L961P, I944N, G1043D) display sensitivity to PARPi treatment in clonogenic survival assays. As negative controls, we included 2 VUS (E1081D, E42K) and 1 benign variant (D134N) that did not impair HR, whereas a truncating variant (Y551) served as a positive control (Fig. 2). The benign variant and 2 VUS that did not affect HR also did not render cells sensitive to PARPi treatment. In contrast, the 4 novel VUS and the truncating variant rendered cells hypersensitive to PARPi treatment to a similar extent as observed for PALB2 KO cells expressing Ev (**see new results in Fig. 3c**). These results are highly consistent with those from the proliferation based PARPi sensitivity assay (Fig. 3a).

Reviewer #2 (Remarks to the Author):

This interesting paper reports the development of an assay system to evaluate the function of mis-sense mutations in the cancer suppressor PALB2, and its use to show that certain PALB2 variants of previously uncertain pathogenic significance (VUSs) affect aspects of biochemical or cellular function.

To develop their assay system, the authors integrate the well-established DR-GFP reporter for measuring proficiency in homologous recombination (HR) into the genome of murine embryonic stem (mES) cells, then delete endogenous murine p53 and Palb2 genes, while introducing various forms of human PALB2 cDNA into the Rosa26 locus. The functional effect of 48 different PALB2 VUSs on HR was then examined, leading to the determination that several different VUSs are potentially deleterious to PALB2 function. Further functional analyses were carried out to validate that these VUSs affect known PALB2 functions relevant to HR.

The strengths of the paper are that it is (generally) technically sound and well written. The major weakness is that the work represents a limited scientific and technical advance, which is more suited to a specialist journal. In addition, several factors limit the potential significance and impact of the work.

Conceptually similar assays to determine the functional effects of VUSs have already been reported for related proteins like BRCA1 and BRCA2 (eg., Chang et al J Clin Invest 2009; Biswas et al Hum Mol Genet 2012; Guidugli et al Hum Mutat 2014; Guidugli et al Am J Hum Genet 2018). In addition, more recent work reports conceptually novel approaches using saturation genome editing to address the assignment of VUSs in BRCA1 (eg., Findlay et al Nature 2018). Placed in the context of this prior literature, the manuscript represents a somewhat limited scientific and technical advance.

The novel approach using saturation genome editing to address the assignment of VUS in BRCA1 (e.g. Findlay *et al.*, Nature, 2018) is certainly the stepping stone towards similar studies of VUS in BRCA2 and PALB2 (as also acknowledged in the discussion of our manuscript). Here we have setup a validated assay for the functional classification of variants in PALB2. This assay will not only allow us to estimate cancer risk associated with VUS in PALB2, which is important for clinical utility, but may also be employed for the high-throughput analysis of VUS in PALB2. Indeed, by using libraries of thousands of variants in PALB2 we have begun such an analysis. However, this study is far from being completed and is beyond the scope of the current paper.

PALB2 has been implicated in cellular functions other than HR (eg., Simhadri et al., Oncogene 2019; Ma et al MCB 2012) that are plausibly relevant to cancer suppression. It remains possible that VUSs that do not score in the HR assays reported in this paper may nevertheless be deleterious.

Ma *et al.*, (MCB, 2012) reported that PALB2 plays a role in controlling the reactive oxygen species (ROS) levels in human cells. In this study, ROS was measured by using the non-fluorescent compound 2',7'-dichlorofluorescein diacetate (DCFDA), which upon cellular uptake can be oxidized by ROS into 2', 7' -dichlorofluorescein (DCF). DCF is a highly fluorescent compound which can be detected by flow cytometry. Using this approach, they found that ROS levels were increased in human PALB2 knockdown cells. Consistently, we observed increased ROS levels in our PALB2 KO mES cells complemented with an empty vector (Ev) compared to the same cells complemented with wild-type-type PALB2 (**reviewer only Figure 1RO**). However, given that we measured these samples with a more than 5 minutes time interval at the flow cytometer with the Ev-complemented cells being measured first, we wondered whether diffusion of DCF out of the PALB2 KO cells expressing wild-type-type PALB2 during this interval, could explain the observed difference in ROS levels. Indeed, when we measured ROS levels in PALB2 KO cells expressing wild-type-type PALB2, we observed a time-dependent decrease in the ROS levels, indicating that the time between measurements of different samples influences the outcome in this assay (**reviewer only Figure 1RO**). To overcome this problem, we changed our experimental setup to a 96-well format, allowing rapid flow cytometry-based measurements of samples with time intervals <1 minute. Strikingly, however, by using this approach, we did not detect a significant difference in ROS levels between PALB2 KO mES cells expressing an empty vector (Ev) when compared to the same cells expressing wild-type-type type PALB2 (**reviewer only Figure 1RO**). Also,

when we included 1 benign, 3 truncating and 4 VUS, one of which did not affect HR (E42K), we did not detect any significant change in ROS levels between cells expressing wild-type type PALB2 and PALB2 with either of these variants (**reviewer only Figure 1RO**). Together this suggest that the PALB2-deficient cells show only a minor change in ROS levels, if at all, which does not warrant the testing of variants in PALB2 for their effect on ROS.

Reviewer only Figure 1RO a ROS levels as measured in *Trp53^{KO}/Palb2^{KO}* mES cells expressing wild-typetype (WT) human *PALB2* or empty vector (Ev) with the indicated time interval at which the samples were analyzed by flow cytometry. **b** As in **a**, except that only *Trp53^{KO}/Palb2^{KO}* mES cells expressing wild-typetype (WT) human *PALB2* were analyzed with the indicated time intervals. **c** As in **a**, except that *Trp53^{KO}/Palb2^{KO}* mES cells expressing the indicated variants were analyzed in 96-well format, allowing sample analysis with short(er) time intervals (<1 minute).

Simhadri *et al.* (Oncogene, 2019) reported that PALB2 plays a role in the activation and/or maintenance of the DNA-damage-induced G2/M checkpoint in mouse and human cancer cells. We exposed wild type and PALB2 KO mES cells to 3 or 10 Gy ionizing radiation and 1 and 6 hours later immune-stained the cells with anti-H3-S10-phospho to measure the fraction M-phase cells by flow cytometry, which would reveal whether a larger fraction of cells had

moved from G2 into M-phase either due to a defect in the activation (measured at 1 hour) or maintenance (measured at 6 hours) of the G2/M checkpoint. In accordance with Simhadri *et al.*, Oncogene (2019), we found that the loss of PALB2 did not affect G2/M checkpoint activation, but strongly impaired G2/M checkpoint maintenance (**see new results in Fig. 6a**), a phenotype resembling that of human cancer cells depleted of PALB2. Next, we assessed the effect of the arbitrarily selected 11 VUS (Y1064C, S1058P, L931R, E1018D, P707L, Y408H, E42K, D871G, K18R L939W, R37H) that did not grossly affect HR in DR-GFP and PARPi sensitivity assays (Figs. 2B and 3A). As negative controls two benign variants (D134N, G998E) that were HR-proficient were included, while two truncating (Y551X, Y1183X), four VUS (A1025R, G1043D, L961P, L35P) and an empty vector (Ev) that were HR-deficient served as positive controls (Figs. 2B and 3A) (see also reviewer 1, point 1). The A1025A variant was included as it was shown to impair the PALB2-BRCA2 interaction without affecting PALB2 stability/expression (2). As expected and in agreement with a previous report (2), we found that the expression of this variant rendered cells defective in HR and hypersensitive to PARPi treatment (**see new results added to Figs. 2b and 3a**). Strikingly, the 2 benign variants and 11 VUS that did not grossly impact HR, did not impact G2/M checkpoint maintenance when compared to cells expressing wild-type (WT) PALB2 (**see new results in Fig. 6b**). In contrast, the 2 truncating variants and 4 VUS that impaired HR, also impaired the maintenance of G2/M arrest comparable to that observed for PALB2 KO cells with Ev (**see new results in Fig. 6b**). The fact that L35P and A1025R, which impair the PALB2-BRCA1 and PALB2-BRCA2 interactions, respectively, both impair G2/M checkpoint maintenance, suggests that PALB2 cooperates with BRCA1 and BRCA2 in this process, which is consistent with previous observations (3). Importantly, when comparing the G2/M checkpoint maintenance efficiency and HR efficiency measured by DR-GFP assays, a strong negative correlation was observed for all variants tested ($R^2=0.8577$) (**see new results in Fig. 6c**). Thus, VUSs that do not score as deleterious in the HR assays also do not score as such in this checkpoint assay.

Finally, characterization of the potential mechanism of VUSs identified here remains preliminary. For example, p.R37H in the PALB2 N-terminus impairs HR but apparently does not affect the BRCA1-PALB2 interaction. Much further work is necessary to establish how this mutation may work to affect PALB2 function.

R37 is located in the CC domain of PALB2, which is essential for PALB2's interaction with BRCA1, suggesting that R37H may impact this interaction. However, Foo *et al.* (Oncogene, 2017) reported that R37H does not affect the PALB2-BRCA1 interaction. This was somewhat surprising given that the authors also showed that this variant partially impairs HR, which is consistent with our observations (Fig. 2b). We therefore decided to assess ourselves whether R37H would affect the PALB2-BRCA1. GFP-NLS, YFP-PALB2 and YFP-PALB2-R37H were expressed in U2OS cells and subjected to GFP/YFP pulldowns followed by western blot analysis of endogenous BRCA1. This revealed that YFP-PALB2-R37H, shows a reduced ability to interact with BRCA1 when compared to YFP-PALB2 (**new results in Fig. 5b**). Thus, R37H, similar to L24S and L35P, affects the PALB2-BRCA1 interaction. Given that the BRCA1-PALB2 interaction is required for PALB2 localization at sites of DNA damage (4), we expressed YFP-PALB2, YFP-PALB2-L24S, YFP-PALB2-R37H and YFP-PALB2-L35P in U2OS cells and assessed their accumulation at DNA damage-containing tracks generated by laser micro-irradiation. In agreement with a previous report (4), we observed that L35P impairs the recruitment of PALB2 to DNA damage sites when compared to that of wild-type type

PALB2. Moreover, we also found that L24S and R37H impaired PALB2 recruitment to sites of DNA damage (**new results in Fig. 5c,d**). Thus, L24S and R37H affect PALB2 accumulation at DNA breaks by reducing the binding to BRCA1, explaining the reduced HR efficiency in cells expressing these variants (Figs. 2b and 3c) (Foo *et al.*, Oncogene, 2017).

Reviewer #3 (Remarks to the Author):

The authors describe an elegant in vivo assay that allows the effects of single-nucleotide variants on the function of the HR scaffold protein PALB2 to be examined.

They describe several sets of experiments which look at Variants of Unknown Significance (VUS), their effects on the homologous recombination pathway of DNA repair and the sensitization of cells to PARP inhibition. They also go onto validate their results for selected VUS in human cells.

Overall the manuscript is of a high standard, and should be considered for publication after some minor revision.

MAJOR POINTS:

> Panel C of Figure 4 is very difficult to interpret or follow — and actually only shows the environment of the native amino acid, and not the predicted effect of the SNP / VUS — for a reader well-versed in structural biology this does not represent a particular issue, but a more general audience would find this problematic. This figure should be revisited and revised. A key should also be provided alongside the figure to aid the reader.

To better visualize the structural effects of the different VUS, we now show structures with the native amino acid, as well as with the corresponding VUS side by side (**see new results in Fig. 4c and Supplementary Fig. 10**). We also visualize the loss of particular interactions (hydrophobic interaction, ionic interaction and/or hydrogen bond) in the structures containing a VUS. A legend is now provided alongside the figure to aid the reader (**see new results in Fig. 4c and Supplementary Fig. 10**).

> The manuscript also describes each the predicted structural effects of each VUS in a different order to that presented in the figure, again potentially affecting the manuscript's readability and ease of interpretation.

We now discuss the predicted structural effects of each VUS in the text in the order to that presented in the figure (**see new results in Fig. 4c and Supplementary Figure 10**).

Page 12: However, it is important to note that in the set of PALB2 missense variants analyzed in this study, there are several variants for which similar destabilizing effects would be predicted by using in silico modeling, but which we found to be (completely) functional in our HR-assays

> Which missense variants are these? This currently reads a little like a throw-away unsupported statement.

We have indicated in the text the different PALB2 variants (p.L931R, p.E1018D, p.D871G and p.W1164C) for which in silico modeling predicted protein destabilizing effects, while retaining protein function as measured by their effect on HR.

Page 6; semi high-throughput

> Can an assay be described as 'semi' high-throughput? Consider changing to medium through-put.

This seems a semantic issue. We prefer “semi high-through” over “medium through-put”.

Page 6; HR is highly efficient in this phase of the cell cycle

>HR in S-phase is actually reasonably restricted, as HR cannot occur until a sister chromatid template becomes available — which itself is determined by when a particular stretch of DNA is replicated, and hence to whether it is in proximity to an early or late-firing origin. Please consider rephrasing accordingly.

We have changed the text to “HR becomes active in this phase of the cell cycle”.

References

1. Michl J, Zimmer J, Tarsounas M. Interplay between Fanconi anemia and homologous recombination pathways in genome integrity. *The EMBO journal* **2016**;35(9):909-23 doi 10.15252/embj.201693860.
2. Oliver AW, Swift S, Lord CJ, Ashworth A, Pearl LH. Structural basis for recruitment of BRCA2 by PALB2. *EMBO reports* **2009**;10(9):990-6 doi 10.1038/embor.2009.126.
3. Simhadri S, Vincelli G, Huo Y, Misenko S, Foo TK, Ahlskog J, *et al.* PALB2 connects BRCA1 and BRCA2 in the G2/M checkpoint response. *Oncogene* **2019**;38(10):1585-96 doi 10.1038/s41388-018-0535-2.
4. Foo TK, Tischkowitz M, Simhadri S, Boshari T, Zayed N, Burke KA, *et al.* Compromised BRCA1-PALB2 interaction is associated with breast cancer risk. *Oncogene* **2017**;36(29):4161-70 doi 10.1038/onc.2017.46.

REVIEWERS' COMMENTS:

Reviewer #1 (Remarks to the Author):

The authors have adequately addressed my concerns.

Reviewer #2 (Remarks to the Author):

The authors have added new experimental data in the revised paper to address my original comments. These changes have improved the manuscript and in part addressed my concerns, as follows.

One original concern was that this work, although technically sound, represents a limited technical advance in light of new approaches like saturation genome editing, which have already been deployed to address in a more comprehensive manner how known variants of unknown significance in related genes like BRCA1 affect function.

The author's response acknowledges this issue, but consider its resolution beyond the scope of the present work. They also say that work is already in progress to provide a more comprehensive analysis of PALB2 variants.

A second concern was the limited scientific and conceptual insight afforded by this work into the mechanism of PALB2 function. Here, the authors provide in the revised paper new results to show that the deleterious PALB2 variants that they have identified do not affect cellular ROS accumulation, but do affect G2 checkpoint maintenance and PALB2 recruitment to DNA breaks by reducing BRCA1 interaction. While technically sound, these results do not identify novel mechanisms or insights into PALB2 function.

In summary, while appreciating the nice work carried out by the authors, I remain unconvinced that the manuscript represents a sufficient advance to be published in Nature Comms, but instead, is more suited to a specialist journal. As the authors acknowledge, a more comprehensive analysis of the effect of PALB2 variants is already underway, and could soon supersede the results presented in this manuscript.

Reviewer #3 (Remarks to the Author):

The authors have made a number of amendments and alterations to the manuscript, which act to improve both its clarity and impact.

I am satisfied with the point-by-point responses to reviewers, and would recommend that the manuscript is now suitable for publication.

I would suggest however, that the authors take care to differentiate the A1205R mutation in PALB2 from the other VUS described within; to the best of my knowledge this is a specifically designed mutation rather than one found in patients per se — I will leave this potential amendment to the editorial team's discretion.

Point-by-point response to reviewer comments

NCOMMS-19-10676B

Functional analysis of genetic variants in the high-risk breast cancer susceptibility gene PALB2

Rick A.C.M. Boonen, Amélie Rodrigue, Chantal Stoepker, Wouter W. Wiegant, Bas Vroling, Milan Sharma, Magda Rother, Nandi Celosse, Maaïke P.G. Vreeswijk, Fergus Couch, Jacques Simard, Peter Devilee, Jean-Yves Masson and Haico van Attikum

We would like to thank the reviewers for their positive feedback. We have addressed the remaining constructive comment of Reviewer #3 as shown below (text in red highlights our response).

1. REVIEWER COMMENTS

Reviewer #1 (Remarks to the Author):

The authors have adequately addressed my concerns.

Reviewer #2 (Remarks to the Author):

The authors have added new experimental data in the revised paper to address my original comments. These changes have improved the manuscript and in part addressed my concerns, as follows.

One original concern was that this work, although technically sound, represents a limited technical advance in light of new approaches like saturation genome editing, which have already been deployed to address in a more comprehensive manner how known variants of unknown significance in related genes like BRCA1 affect function.

The author's response acknowledges this issue, but consider its resolution beyond the scope of the present work. They also say that work is already in progress to provide a more comprehensive analysis of PALB2 variants.

A second concern was the limited scientific and conceptual insight afforded by this work into the mechanism of PALB2 function. Here, the authors provide in the revised paper new results to show that the deleterious PALB2 variants that they have identified do not affect cellular ROS accumulation, but do affect G2 checkpoint maintenance and PALB2 recruitment to DNA breaks by reducing BRCA1 interaction. While technically sound, these results do not identify novel mechanisms or insights into PALB2 function.

In summary, while appreciating the nice work carried out by the authors, I remain unconvinced that the manuscript represents a sufficient advance to be published in Nature Comms, but instead, is more suited to a specialist journal. As the authors acknowledge, a more comprehensive analysis of the effect of PALB2 variants is already underway, and could soon supersede the results presented in this manuscript.

Reviewer #3 (Remarks to the Author):

The authors have made a number of amendments and alterations to the manuscript, which act to improve both its clarity and impact.

I am satisfied with the point-by-point responses to reviewers, and would recommend that the manuscript is now suitable for publication.

I would suggest however, that the authors take care to differentiate the A1205R mutation in PALB2 from the other VUS described within; to the best of my knowledge this is a specifically designed mutation rather than one found in patients per se — I will leave this potential amendment to the editorial team's discretion.

Revise your paper one last time to address the remaining concerns of Reviewer #3 who has asked that you clarify within your manuscript that the A1205R mutation in PALB2 is not tumour-derived but rather is a synthetic mutation.

In the manuscript we now clearly distinguish p.A1025R as a synthetic missense variant (page 7) and throughout the manuscript we no longer discuss it as a VUS. Furthermore, we have changed Figures 2, 3 and 6 by giving p.A1025R a separate color (purple). The corresponding figure legends have been changed accordingly as well.